# Regional associations between inspiratory tongue dilatory movement and genioglossus activity during wakefulness in people with obstructive sleep apnoea

Lauriane Jugé[1,2], Angela Liao[1,2], Jade Yeung[1] (iD), Fiona L. Knapman[1,2] (iD), Christopher Bull[1,2] (iD), Peter G.R. Burke[1,3], Elizabeth C. Brown[1,4], Simon C. Gandevia[1,2] (iD), Danny J. Eckert[1,2,5] (iD), Jane E. Butler[1,2] (iD) and Lynne E. Bilston[1,2] (iD)

[1] *Neuroscience Research Australia, Sydney, New South Wales, Australia*
[2] *Faculty of Medicine and Health, University of New South Wales, Sydney, New South Wales, Australia*
[3] *Macquarie Medical School, Faculty of Medicine and Health Sciences, Macquarie University, Sydney, New South Wales, Australia*
[4] *Prince of Wales Hospital, Sydney, New South Wales, Australia*
[5] *Adelaide Institute for Sleep Health and Flinders Health and Medical Research Institute, Flinders University, Adelaide, Australia*

Handling Editors: Scott Powers & Frank Powell

The peer review history is available in the Supporting Information section of this article (https://doi.org/10.1113/JP285187#support-information-section).

Regional associations between inspiratory tongue dilatory movement and genioglossus activity during wakefulness in people with obstructive sleep apnoea (OSA)

**Abstract**  Inspiratory tongue dilatory movement is believed to be mediated via changes in neural drive to genioglossus. However, this has not been studied during quiet breathing in humans. Therefore, this study investigated this relationship and its potential role in obstructive sleep apnoea (OSA). During awake supine quiet nasal breathing, inspiratory tongue dilatory movement, quantified with tagged magnetic resonance imaging, and inspiratory phasic genioglossus EMG

normalised to maximum EMG were measured in nine controls [apnoea–hypopnea index (AHI) ≤5 events/h] and 37 people with untreated OSA (AHI >5 events/h). Measurements were obtained for 156 neuromuscular compartments (85%). Analysis was adjusted for nadir epiglottic pressure during inspiration. Only for 106 compartments (68%) was a larger anterior (dilatory) movement associated with a higher phasic EMG [mixed linear regression, beta = 0.089, 95% CI [0.000, 0.178], $t(99) = 1.995$, $P = 0.049$, hereafter EMG↗/mvt↗]. For the remaining 50 (32%) compartments, a larger dilatory movement was associated with a lower phasic EMG [mixed linear regression, beta = −0.123, 95% CI [−0.224, −0.022], $t(43) = −2.458$, $P = 0.018$, hereafter EMG↘/mvt↗]. OSA participants had a higher odds of having at least one decoupled EMG↘/mvt↗ compartment (binary logistic regression, odds ratio [95% CI]: 7.53 [1.19, 47.47] ($P = 0.032$). Dilatory tongue movement was minimal (>1 mm) in nearly all participants with only EMG↗/mvt↗ compartments (86%, 18/21). These results demonstrate that upper airway dilatory mechanics cannot be predicted from genioglossus EMG, particularly in people with OSA. Tongue movement associated with minimal genioglossus activity suggests co-activation of other airway dilator muscles.

(Received 21 June 2023; accepted after revision 20 October 2023; first published online 20 November 2023)

**Corresponding author** Lynne E. Bilston: Neuroscience Research Australia, Margarete Ainsworth Building, Barker Street Randwick, NSW 2031, Australia. Email: l.bilston@neura.edu.au

**Abstract figure legend** The relationship between regional inspiratory tongue movement and genioglossus activity in people with and without obstructive sleep apnoea (OSA) is examined in this study during awake supine quiet nasal breathing. Our study showed that higher genioglossus phasic EMG does not consistently translate into tongue dilatory movement. Large dilatory tongue movements can occur despite minimal genioglossus inspiratory activity, particularly in people with OSA, despite them having a compromised airway more and not less negative nadir epiglottic pressure.

## Key points

- Inspiratory tongue movement is thought to be mediated through changes in genioglossus activity. However, it is unknown if this relationship is altered by obstructive sleep apnoea (OSA).
- During awake supine quiet nasal breathing, inspiratory tongue movement, quantified with tagged magnetic resonance imaging (MRI), and inspiratory phasic genioglossus EMG normalised to maximum EMG were measured in four tongue compartments of people with and without OSA.
- Larger tongue anterior (dilatory) movement was associated with higher phasic genioglossus EMG for 68% of compartments.
- OSA participants had an ∼7-times higher odds of having at least one compartment for which a larger anterior tongue movement was *not* associated with a higher phasic EMG than controls. Therefore, higher genioglossus phasic EMG does not consistently translate into tongue dilatory movement, particularly in people with OSA.
- Large dilatory tongue movements can occur despite minimal genioglossus inspiratory activity, suggesting co-activation of other pharyngeal muscles.

**Lauriane Jugé** is a Senior Research Fellow in biomedical imaging at Neuroscience Research Australia (NeuRA) and conjoint Lecturer at the Faculty of Medicine of the University of New South Wales. She conducts pioneering multidisciplinary research studies using multimodal imaging techniques to generate novel mechanistic insights into physiological problems, such as sleep disorders and neural injury. Her expertise covers the broad scope of basic to clinical science. **Angela Liao** is a junior doctor (BMed/MD) at Concord Repatriation General Hospital, Sydney, Australia, and an adjunct associate lecturer for the School of Medicine, University of New South Wales. She is currently pursuing further studies in health data science. Angela is interested in radiology and applying novel technologies to optimise patient outcomes.

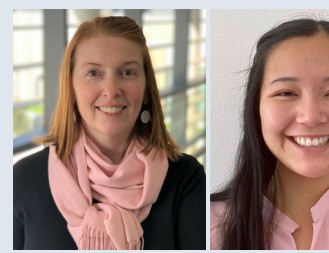

## Introduction

Upper airway muscle behaviour in tasks such as swallowing, speech and breathing is governed by the coordination of complex neuromuscular systems. The soft tissues of the upper airway are highly deformable and airway shape is a result of the combined action of the surrounding dilator and constrictor muscles. In particular, negative inspiratory intraluminal pressure due to craniofacial variations or weight gain that increase extraluminal pressure increase susceptibility to upper airway collapse (Gleadhill et al., 1991). Failure to sufficiently recruit and coordinate dilator muscles results in repeated apnoeas and/or airflow limitation during sleep, as occurs in obstructive sleep apnoea (OSA) (Edwards & White, 2011; Strohl et al., 2012). OSA affects at least 10–20% of adults (Lechat et al., 2022) and is becoming more common with an ageing population and increasing obesity rates (Motamedi et al., 2009; Peppard et al., 2013). The disorder produces increased sleepiness and is linked to cardiovascular disease (Gottlieb et al., 2010; Somers et al., 2008), decreased quality of life (Baldwin et al., 2001; Peppard et al., 2006) and cognitive function (Kim et al., 1997; Yaffe et al., 2011).

Effective inspiratory tongue dilatory function is critical to maintain airway patency. As upper airway collapse occurs rarely during wakefulness, reduced neural drive to the tongue and upper airway muscle reflexes in people with OSA during sleep are thought to contribute to the inability of the tongue to maintain airway patency (e.g. Carberry et al., 2022; Horner et al., 1994). Furthermore, stimulation of the genioglossus, the largest dilator muscle of the upper airway (Takemoto, 2001), reduces upper airway collapsibility in OSA (Cori et al., 2018), and some obese people with a compromised upper airway do not have OSA due to enhanced dilator muscle activity during sleep (Sands et al., 2014). However, it is unclear why some are unable to activate their tongue muscles to maintain airway patency during sleep (Gell et al., 2022, Younes, 2008). Understanding the mechanisms underlying effective upper-airway muscle responses is essential to target OSA treatments such as neural stimulation or pharmacological therapies that alter muscle activity.

Tongue muscle activity is driven by phasic and tonic inputs from the brainstem respiratory control centres and local reflexes triggered by negative airway pressure during inspiration (Vranish & Bailey, 2015). Neural drive increases during inspiration and decreases during expiration. This correlates with tongue contraction and forward movement during inspiration and relaxation posteriorly during expiration in awake healthy adults (Cheng et al., 2008). Anterior movement of the posterior tongue during inspiration dilates the airway (Cheng et al., 2008; Kwan et al., 2019), and this movement starts before inspiratory flow to prepare the airway for inspiration (Cheng et al., 2011). During wakefulness, drive from the central pattern generator to the genioglossus begins ∼100 ms before the onset of inspiration (Butler, 2007; Strohl et al., 1980), while drive from the reflex mechanism occurs ∼20 ms after the onset of inspiration (Carberry et al., 2015; Eckert et al., 2007). Still, during wakefulness, it has been reported that both phasic inspiratory and tonic components of genioglossus EMG may be higher in people with OSA than in healthy control participants (Fogel et al., 2001), possibly to counteract the increased negative pressure in the upper airway during inspiration in people with OSA. Inspiratory airway dilatation during wakefulness is larger in those with narrower airways (Cheng et al., 2014) and with OSA (Juge, Knapman et al., 2020).

Regional variation in inspiratory tongue dilatory movement has also been reported in awake people with and without OSA (Brown et al., 2013; Cai et al., 2016; Juge, Knapman et al., 2020; Juge, Yeung et al., 2020; Juge et al., 2021), with more movement likely to occur in the posterior section of the horizontal compartment of the tongue (Cheng et al., 2008; Juge, Knapman et al., 2020). The horizontal and oblique neuromuscular compartments of the genioglossus are innervated by different hypoglossal nerve branches (Mu & Sanders, 2010) and have different muscle fibre arrangements (Luu et al., 2018; Sanders & Mu, 2013), supporting functional compartmentalisation of the tongue (Bilston & Gandevia, 2014). Regional variation in EMG to the tongue has also been reported in awake healthy adults during breathing, although the results were conflicting with one study reporting larger activity in the posterior genioglossus than in the anterior region (Vranish & Bailey, 2015), and the other reporting more EMG anteriorly (Eastwood et al., 2003).

These previous studies suggest that inspiratory tongue dilatory movement is the biomechanical result of neural drive to the genioglossus. However, comprehensive studies of the relationship between inspiratory genioglossus neural drive and tongue dilatory movement during quiet breathing have not been conducted and whether this relationship is altered in OSA is unknown. Thus, this study aimed to examine the relationship between regional inspiratory tongue movement, as measured by tagged magnetic resonance imaging (MRI), and genioglossus activity, as quantified by intramuscular EMG. A secondary aim was to investigate how this relationship between inspiratory tongue movement and genioglossus activity is related to OSA pathophysiology. We hypothesised that higher genioglossus EMG would be associated with larger dilatory tongue movement of a given neuromuscular compartment. We also hypothesised that decoupling between genioglossus neuromuscular compartmental activity and regional tongue movement would occur more commonly in people with OSA compared to healthy controls as regional genioglossus motor unit size increases in people with OSA (Saboisky

et al., 2007, 2012, 2015) could cause a discrepancy between the mechanical response of the tongue and the EMG.

## Methods

### Ethical approval

The study complies with the *Declaration of Helsinki* (2013), except for database registration, and was approved by the South Eastern Sydney Local Health District Human Research Ethics Committee (HREC/13/POWH/745). All participants provided written informed consent.

### Participants

Sixty-three participants were recruited. Standard diagnostic polysomnography was conducted within a year of the recruitment to determine the apnoea–hypopnoea index (AHI) [American Academy of Sleep Medicine criteria v2.4 (3% desaturation) (Berry et al., 2017)], except for three participants who had their polysomnography study between 1 and 2.4 years from recruitment but did not report weight change during this period. For consistency, all polysomnography not conducted at one of two main sites involved in the study (Neuroscience Research Australia and Royal North Shore

Hospital, Australia) were rescored. Participants were categorised as having OSA (AHI > 5 events/h) or no OSA (AHI ≤ 5 events/h). People with a history of upper airway surgery or chronic illnesses that could impact respiratory muscle function and upper airway activity/drive, or those taking medications, including sedatives and psychoactive medications that might interfere with the outcomes of the research study were excluded.

### Magnetic resonance imaging

Participants were imaged using an Achieva 3TX MRI (Philips Healthcare, Best, The Netherlands) and a neurovascular coil during nasal breathing in wakefulness while lying supine. The detailed imaging and analysis protocols have been described previously (Juge, Knapman et al., 2020). In brief, we collected axial anatomical images with a turbo spin echo sequence to measure the nasopharyngeal upper airway cross-sectional area (behind the soft palate), where it is narrowest (Fig. 1*A*), and sagittal anatomical images (also with a turbo spin echo sequence) to measure the soft palate length, defined as the linear distance between its origin at the hard palate and the caudal tip of the uvula, using ImageJ 1.51n (NIH, Bethesda, MD, USA) (Fig. 1*B*). A three-dimensional, multi-echo 2-Point Dixon (mDIXON)

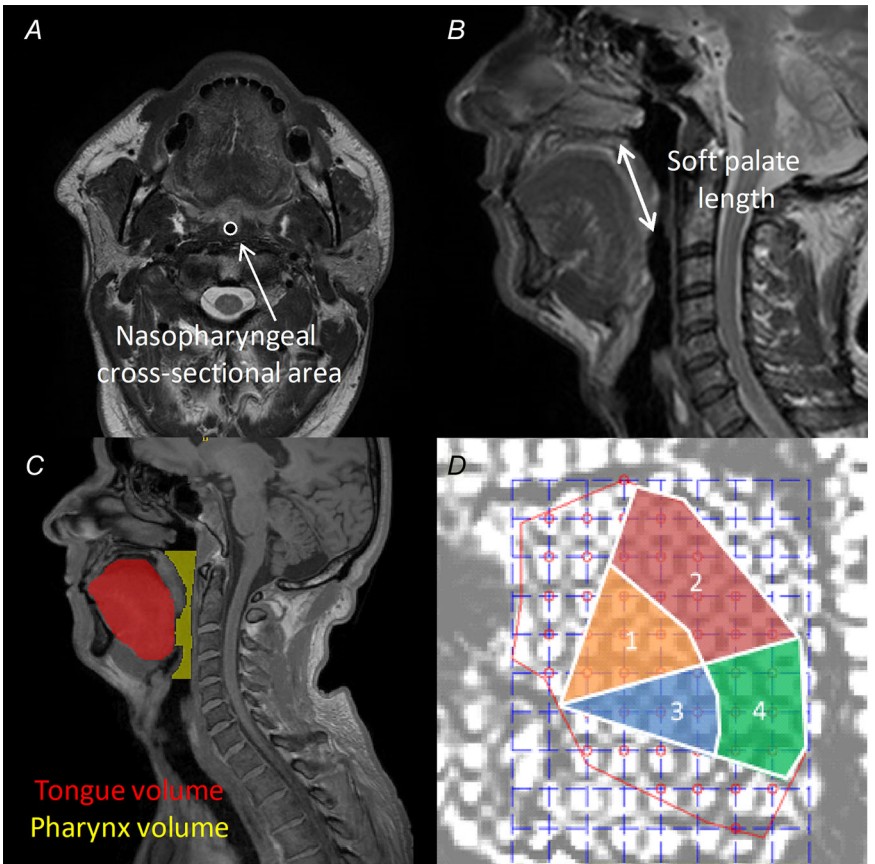

**Figure 1. Typical anatomical, tagged MRI, and mDIXON images collected, and neuromuscular tongue compartments**
*A*, axial anatomical image. The nasopharyngeal cross-sectional area is indicated by the white arrow. *B*, sagittal anatomical image. The soft palate length was defined by the linear distance between its origin at the hard palate and the caudal tip of the uvula. *C*, water + fat images (mDIXON). The tongue volume (red) included the horizontal and oblique compartments and the tip of the tongue. The pharyngeal volume (yellow) was measured from where the soft palate originates to the base of the epiglottis. *D*, tagged MRI images. Tongue inspiratory dilatory function, as measured as the anterior–posterior movement, was quantified in four neuromuscular compartments: (1) anterior oblique, (2) posterior oblique, (3) anterior horizontal and (4) posterior horizontal. [Colour figure can be viewed at wileyonlinelibrary.com]

scan was also acquired to obtain pharynx and whole tongue volumetric measurements from the water + fat images using ITK-SNAP (version 3.6.0, USA) (Yushkevich et al., 2006) (Fig. 1*C*). Mid-sagittal tagged MRI sequences were acquired with a modified complementary spatial modulation of magnetisation sequence to track and quantify inspiratory tongue movement during inspiration by superimposing a grid of altered magnetisation on the tissues (Axel & Dougherty, 1989; Brown et al., 2013; Cheng et al., 2014). A respiratory sensor MR-compatible device (Philips Healthcare, Best, The Netherlands) that can capture respiratory motion with an inflated cushion positioned just below the sternum was used to align the tagged images with the respiratory cycle.

The following imaging parameters were used for the anatomical axial images: repetition time 4835 ms, echo time 80 ms, 50 continuous axial slices of 3 mm thickness, field of view 192 × 192 mm, in-plane spatial resolution $0.50 \times 0.50$ mm$^2$, total scan duration 4:21 min. The sagittal anatomical acquisition was collected with imaging parameters: repetition time 3.0 s, echo time 52 ms, 25 continuous sagittal slices of 3 mm, field of view 192 × 192 mm, in-plane spatial resolution $0.75 \times 0.75$ mm$^2$, total scan duration 1:48 min. For the mDIXON scan, the imaging parameters were: repetition time 4.6 ms, first echo time 1.45 ms, second echo time 2.6 ms, 150 continuous sagittal slices of 1 mm, field of view 300 × 300 mm, in-plane spatial resolution $0.94 \times 0.94$ mm$^2$ and total scan duration 3:31 min. For the tagged MRI scans, sagittal images were acquired every 250 ms for 30 s to record up to 10 breaths. The imaging parameters used were: flip angle 7°, repetition time 2.2 ms, echo time 0.9 ms, one mid-sagittal slice of 10 mm, field of view 220 × 220 mm, in-plane spatial resolution $0.86 \times 0.86$ mm$^2$ and tag spacing of 7.21 mm. Five tagged MRI scans were collected.

Sagittal anterior dilatory movements of the tongue from the onset of inspiration to the onset of expiration were measured perpendicular to the posterior pharynx wall (measured on the sagittal anatomical images) using harmonic phase methods (Osman et al., 1999) implemented in Matlab (The MathWorks Inc., Natick, MA, USA). This method has a resolution of 0.1 pixels (equivalent to $\sim$ 0.09 mm) (Kerwin & Prince, 2000). All movements were averaged over three inspirations for each participant. A mesh of points with an 8 mm vertical and horizontal spacing was used to quantify inspiratory tongue movement in the following neuro-muscular compartments of the tongue: (1) anterior oblique, (2) posterior oblique, (3) anterior horizontal and (4) posterior horizontal (Fig. 1*D*). Positive displacements represent anterior (dilatory) movement.

Anterior dilatory movements were classified as minimal if less than 1 mm and large if greater than 1 mm, as previously defined (Brown et al., 2013).

Previous definitions (Juge, Knapman et al., 2020) of four inspiratory dilatory patterns were used to classify regional tongue movement: (1) *Minimal* (anterior movement of all four neuromuscular compartments >1 mm); (2) *En bloc* (anterior movement of all four neuromuscular compartments <1 mm); (3) *Oropharyngeal* (anterior movement of the horizontal posterior compartment was <1 mm and the oblique posterior compartments >1 mm); and (4) *Bidirectional* (the two posterior compartments moved in opposite directions).

### Electromyography

EMG was recorded in awake supine participants during nasal breathing using a monopolar electrode configuration with the reference electrode placed on the forehead. Insertion and recording procedures were described in Yeung et al. (2022). We used four percutaneous fine-wire electrodes (Teflon-coated stainless-steel diameter 0.08 mm, #791500 A-M Systems Inc., Sequim, WA, USA) to target the same four tongue neuromuscular compartments defined for the imaging analysis (Fig. 1*D*). Ultrasound was used to guide placement, following EMG recording guidelines (Besomi et al., 2019). The first electrode targeting the oblique anterior neuromuscular compartment was inserted 10 mm behind the posterior edge of the mandibular symphysis, 3 mm lateral to the midline and 15 mm above the superior margin of the geniohyoid. The second electrode targeting the oblique posterior neuromuscular compartment was inserted 20 mm behind the posterior edge of the mandibular symphysis, 3 mm lateral to the midline and 15 mm above the superior margin of the geniohyoid. The third electrode targeting the horizontal anterior neuromuscular compartment was inserted 10 mm to the posterior edge of the mandibular symphysis, 4 mm lateral to the midline and 5 mm above the superior margin of the geniohyoid. Finally, the fourth electrode targeting the horizontal posterior neuromuscular compartment was inserted 20 mm to the posterior edge of the mandibular symphysis, 4 mm lateral to the midline and 5 mm above the superior margin of the geniohyoid. However, ultrasound does not clearly distinguish the border between the oblique and horizontal compartments. Therefore, each intramuscular wire was assigned to one of the four tongue neuromuscular compartments based on the EMG recorded during 'dry' swallowing, as oblique and horizontal compartments have distinct motoneuronal outputs: monophasic for the horizontal and biphasic for the oblique (Yeung et al., 2022).

EMG data were acquired and analysed with Spike2 v6.17 (Cambridge Electronic Design, Cambridge, UK) using a 16-bit analogue-to-digital converter and amplifier (CED1401 and CED1902, respectively, Cambridge

Electronic Design). EMG was collected at 2000 Hz with a 30–1000 Hz bandpass filter. Signals were rectified and smoothed with a time constant of 100 ms before being averaged for each participant. Simultaneously, we measured epiglottic pressure ($P_{epi}$) at 1000 Hz with a pressure-tipped catheter (MPC500, TX, USA) and airflow at 250 Hz with a nasal mask (ResMed, NSW, Australia) connected to a heated pneumotach (3700, Hans Rudolph Inc., Kansas City, MO, USA) and a differential pressure transducer (DP15-16, Validyne Engineering Corp, Northridge, CA, USA). The onset of inspiration was identified by a sharply decreasing epiglottic pressure and increased airflow. The end of inspiration/start of expiration was defined as when the airflow and epiglottic pressure returned to pre-onset inspiration values. Automated breath detection was performed using an algorithm implemented in Matlab (Nguyen et al., 2017). For each wire, phasic (peak minus tonic EMG) and tonic EMG were normalised to the maximum of either swallowing or tongue protrusion (largest of at least two maximal tongue protrusions or swallowing, with at least 30 s of rest between efforts). Multi-unit EMG was analysed during quiet breathing during wakefulness with the participant supine. Occasionally, EMG recording exhibited a prominent single motor unit – these instances were excluded from the multi-unit EMG analysis.

EMG data were recorded separately from the MRI data, as the EMG equipment and electrodes were not MRI-compatible.

## Statistical analysis

All imaging and EMG data were analysed blinded to OSA severity. Imaging analysis was performed blinded to EMG analysis and vice versa. Data are reported as mean ± SD [minimum–maximum]. Statistical analysis was conducted using SPSS v28 (IBM Statistics, New York, NY, USA) and GraphPad Prism [v9.5.0 (730), GraphPad Software, LLC, San Diego, CA, USA). *P* values < 0.050 were considered significant.

Mann–Whitney U-tests were used to assess differences in continuous variables between two groups, while Fisher's exact test was used to determine a difference in the proportion of categorical variables between groups. A mixed linear regression was used to assess the effect of OSA, tongue compartments and their interactions (fixed effects) on phasic and tonic EMG (dependent variables) while adjusting for nadir epiglottic pressure. Other mixed linear regression analyses were used to assess (1) whether the maximum EMG used for the normalisation differed between tongue compartments (fixed effect), and (2) the association between neuro-muscular EMG measurements (dependent variables) and epiglottic nadir pressure (fixed effect). Finally, mixed linear regressions were used to assess the relationships between tongue dilatory movement (dependent variable) and EMG measurements (fixed effects), with the nadir epiglottic pressure as a covariate. All mixed linear regression analyses accounted for repeated measures in participants (random effect).

Two clusters of phasic and tonic EMG (hereafter called 'high' and 'low') were identified from a hierarchical cluster analysis of phasic and tonic EMG performed to determine the number of clusters followed by an exploratory *k*-means cluster analysis to partition the recordings into the number of clusters estimated previously. These two clusters were used to investigate the association between genioglossus activity and large (<1 mm) or minimal (>1 mm) regional inspiratory tongue dilatory movement.

To describe the observed relationships between tongue dilatory movement and phasic EMG, we used the following terminology EMG↗/mvt↗ when a higher phasic EMG was associated with a larger tongue dilatory movement for the neuromuscular tongue compartments classified as having either (1) a 'low' activity with minimal movement or (2) 'high' activity with large movement, and EMG↘/mvt↗ when a lower phasic EMG was associated with a larger tongue dilatory movement (i.e. decoupled relationship) for the neuromuscular tongue compartments classified as having (3) a 'low' activity with large movement or (4) a 'high' activity with minimal anterior movement. Simple linear regression with 95% confidence interval further described these relationships on graphs.

A binary logistic regression analysis was performed to assess the likelihood of participants with OSA of having at least one decoupled EMG↘/mvt↗ tongue compartment compared to controls. Another binary logistic regression analysis was performed to assess the likelihood of participants with at least one decoupled EMG↘/mvt↗ compartment to inspire with a not minimal tongue dilatory pattern compared to those with only EMG↗/mvt↗. Finally, a generalised linear mixed model analysis was used to assess the association between the 'decoupling' status in a particular compartment (i.e. EMG↗/mvt↗ or EMG↘/mvt↗) and OSA status while accounting for repeated measures in participants. All those analyses were adjusted for nadir epiglottic pressure.

For clarity, statistical tests are listed with each result.

## Results

### Participant characteristics

Both tagged MRI and EMG data were obtained for 156 neuromuscular compartments from 46 participants (i.e. in 85% of all 184 possible compartments in the 46 participants). Five participants did not tolerate insertion of the epiglottic catheter or EMG wires, eight others did not have MRI data, either due to claustrophobia

**Table 1. Participant characteristics**

|  | Control participants | OSA participants | *P* value |
|---|---|---|---|
| Number (M:F) | 9 (6:3) | 37 (29:8) | 0.664 |
| AHI (events/h) | 2.8 ± 1.9 [0.5–5.0] | 28.5 ± 21.4 [5.6–94.3] | <0.0001 (***) |
| Age (years) | 48 ± 12 [28–68] | 44 ± 12 [20–73] | 0.370 |
| BMI (kg/m$^2$) | 25.0 ± 3.4 [21.2–31.2] | 27.5 ± 3.9 [21.7–40.4] | 0.087 |
| Tongue volume (cm$^3$) | 134 ± 16 [117–155] | 156 ± 24 [110–209] | 0.015 (*) |
| Soft palate length (mm) | 33 ± 4 [29–41] | 37 ± 6 [28–55] | 0.023 (*) |
| Pharyngeal volume (cm$^3$) | 17.5 ± 4.3 [10.7–22.4] | 18.1 ± 7.8 [5.4–41.4] | 0.765 |
| Nasopharyngeal cross-sectional area (mm$^2$) | 56 ± 26 [16–109] | 52 ± 29 [15–139] | 0.526 |
| Nadir epiglottic pressure (cmH$_2$O) | −1.89 ± 0.64 [−2.93 to −1.19] | −3.67 ± 1.85 [−9.70 to −1.19] | 0.0011 (**) |

Results are presented as mean ± SD with the minimum and maximum values in parentheses. The nasopharyngeal cross-sectional area represents averages over the whole respiratory cycle. Statistical differences between groups for continuous variables were assessed using the Mann–Whitney U-test, and the difference in sex was evaluated using Fisher's exact test. Abbreviations: (M) male, (F) female.

or inadequate image quality, three had non-exploitable EMG measurements in all four tongue neuromuscular compartments, and one participant had both MRI data and EMG data missing.

Among the 46 participants with paired data, nine were control participants (AHI ≤ 5 events/h), 13 had mild OSA (5 < AHI ≤ 15 events/h), 11 had moderate OSA (15 < AHI ≤ 30 events/h) and 13 had severe OSA (AHI > 30 events/h). Of the 156 recordings, 27 were obtained in control participants (17%). Fifty-two were obtained in the horizontal posterior (33%), 34 in the horizontal anterior (22%), 25 in the oblique posterior (16%) and 45 in the oblique anterior (29%) neuromuscular compartments of the tongue.

Table 1 summarises participant characteristics. The population was predominantly male (*n* = 35, 76%), middle-aged (45.2 ± 11.6 years), and overweight (27.0 ± 3.9 kg/m$^2$). Sex, age and body mass index (BMI) did not differ significantly between participants with and without OSA. People with OSA had a larger tongue volume and longer soft palate than control participants, while nasopharyngeal cross-sectional area and pharyngeal volume did not differ between groups. During awake spontaneous breathing, the epiglottic nadir pressure was significantly more negative in OSA participants than in control participants.

## Inspiratory EMG to genioglossus

EMG measurements were averaged over 40 ± 22 [11–113] breaths. When adjusting for nadir epiglottic pressure, for phasic and tonic EMG, no significant effects of OSA status and tongue compartments were found, and there was no significant interaction between OSA status and tongue compartment (Fig. 2 and Table 2). Maximum EMG during tongue protrusion or swallowing did not differ between tongue compartments [mixed linear regression, $F(3,152) = 1.545$, $P = 0.205$].

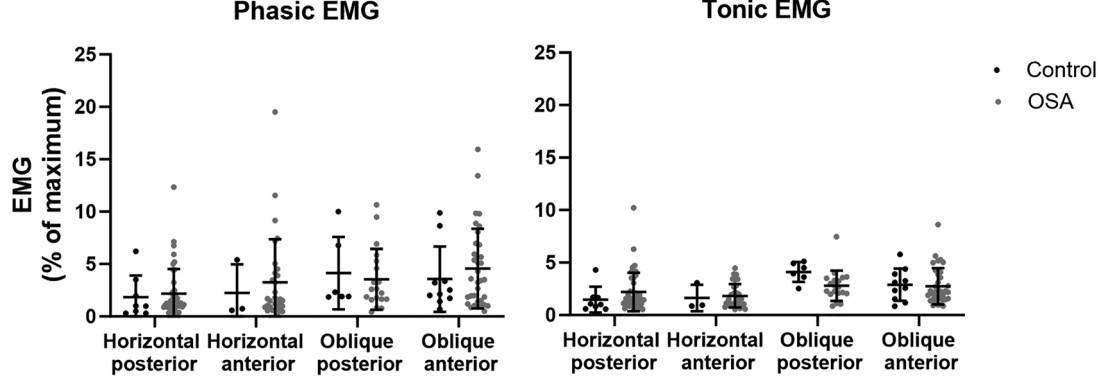

**Figure 2. Phasic and tonic EMG measured in each tongue neuromuscular compartment for control and OSA participants during awake supine quiet inspiration (data points overlaid with mean ± SD)**
Genioglossus phasic and tonic activity did not differ between people with and without OSA and no significant regional variation was observed.

**Table 2. Type III test of fixed effects of the mixed linear regression to determine the effect of OSA, tongue compartments, and their interactions on phasic and tonic EMG**

| Parameter | Phasic EMG[†] | Tonic EMG[†] |
|---|---|---|
| Intercept | $F_{(1,143)} = 7.449$, $P = 0.007$ | $F_{(1,143)} = 15.171$, $P < 0.001$ |
| Tongue compartment | $F_{(3,143)} = 0.251$, $P = 0.861$ | $F_{(3,143)} = 1.284$, $P = 0.282$ |
| OSA status | $F_{(1,143)} = 0.905$, $P = 0.343$ | $F_{(1,143)} = 1.039$, $P = 0.310$ |
| Tongue compartment × OSA status | $F_{(3,143)} = 0.268$, $P = 0.849$ | $F_{(3,143)} = 0.591$, $P = 0.622$ |
| Nadir epiglottic pressure | $F_{(1,143)} = 0.104$, $P = 0.747$ | $F_{(1,143)} = 0.000$, $P = 0.983$ |
| Tongue compartment × nadir epiglottic pressure | $F_{(3,143)} = 0.744$, $P = 0.528$ | $F_{(3,143)} = 0.684$, $P = 0.563$ |
| OSA status × nadir epiglottic pressure | $F_{(1,143)} = 1.173$, $P = 0.281$ | $F_{(1,143)} = 0.528$, $P = 0.469$ |

[†]Dependent variable.
The model accounted for repeated measures in participants (random effect) and the covariate nadir epiglottic pressure (main effects and interactions with OSA status and tongue compartment)

**Table 3. Characteristics of inspiratory genioglossus activity clusters**

| | '*High*'-activity cluster (*n* = 40 wires) | '*Low*'-activity cluster (*n* = 116 wires) | *P* values |
|---|---|---|---|
| Phasic EMG (% of maximum) | 7.91 ± 3.27 [3.23–19.51] | 1.64 ± 1.06 [0.29–4.60] | <0.0001 |
| Tonic EMG (% of maximum) | 4.20 ± 1.73 [1.94–10.22] | 1.76 ± 1.00 [0.50–4.71] | <0.0001 |

Results are presented as mean ± SD with the minimum and maximum values in parentheses. Statistical differences between clusters were assessed using the Mann–Whitney U-test

However, cluster analysis identified clusters of recordings with two levels of genioglossus activity: one small cluster of 40 wires (26%) with a '*high*' activity level, and a larger cluster of 116 wires (74%) with a '*low*' activity level. Phasic and tonic EMG in the compartment in the '*high*' activity cluster were, on average, ∼5- and 2-fold higher, respectively, than that measured in the compartments with '*low*' activity. Table 3 summarises these two levels of inspiratory genioglossus activity.

'*High*' neuromuscular activity was observed in 27 participants (59%, 27/46). When compared to the participants without compartments with '*high*' neuromuscular activity, participants with '*high*' neuromuscular activity had a smaller pharyngeal airway volume (Mann–Whitney U-test, $P = 0.020$, 16.0 ± 6.8 *vs.* 20.7 ± 7.1 cm$^3$), a higher BMI (Mann–Whitney U-test, $P = 0.015$, 28.1 ± 4.3 *vs.* 25.5 ± 2.6 kg/m$^2$) and a smaller nasopharyngeal cross-sectional area (Mann–Whitney U-test, $P = 0.016$, 45.4 ± 27.3 *vs.* 62.6 ± 26.5 mm$^2$). Other participant characteristics did not differ between these groups (Mann–Whitney U-test, AHI $P = 0.795$, age $P = 0.890$, tongue volume $P = 0.546$, soft palate length $P = 0.165$, nadir epiglottic pressure $P = 0.331$; Fisher's exact test, gender proportion $P = 0.092$). The proportion of '*high*' neuromuscular activity did not differ between tongue compartments (Fisher's exact test, $P = 0.122$, horizontal posterior: 17%, 9/52, horizontal anterior: 21%,

7/34, oblique posterior: 28%,7/25, oblique anterior: 29%, 17/45).

Among the 40 compartments with '*high*' activity, a larger negative nadir epiglottic pressure was associated with an increase in phasic, but not tonic, EMG [mixed linear regression, beta = −1.711, 95% confident intervals [−2.792, −0.631], $t_{(38)} = −3.207$, $P = 0.003$, and beta = −0.597, 95% confident intervals [−1.240, 0.046], $t_{(38)} = −1.880$, $P = 0.068$, respectively]. This was not seen for the 116 compartments with '*low*' activity [mixed linear regression, beta = −0.229, 95% confident intervals [−0.600, 0.141], $t_{(114)} = −1.228$, $P = 0.222$, and beta = −0.164, 95% confident intervals [−0.461, 0.133], $t_{(114)} = −1.096$, $P = 0.276$, for the phasic and tonic EMG, respectively; see Fig. 3].

## Regional inspiratory tongue dilatory movement and EMG associations

MRI and EMG experiments were performed an average of 2.5 ± 2.5 [0.0–12.1] weeks apart. Inspiratory tongue movement and patterns have been described previously in controls and OSA participants (Juge, Knapman et al., 2020).

Overall, for regional anterior movement and when adjusting for nadir epiglottic pressure, we found a significant effect of phasic EMG, and two significant

**Table 4. Estimates of fixed effects of the mixed linear regression used to determine the effect of tonic and phasic EMG on inspiratory tongue movement over all tongue compartments**

| Parameter | Estimates of fixed effects over all tongue compartments ($n$ = 156 compartments) |
| --- | --- |
| Intercept | beta = 0.368, 95% CI [−0.077, 0.812], $t$(149) = 1.635, $P$ = 0.104 |
| Tonic EMG | beta = −0.096, 95% CI [−0.196, 0.005], $t$(149) = −1.885, $P$ = 0.061 |
| Phasic EMG | beta = 0.061, 95% CI [0.003, 0.118], $t$(149) = 2.074, $P$ = 0.040 (*) |
| Tonic × phasic EMG | beta = −0.002, 95% CI [−0.007, 0.003], $t$(149) = −0.699, $P$ = 0.486 |
| Nadir epiglottic pressure | beta = 0.091, 95% CI [−0.028, 0.210], $t$(149) = 1.505, $P$ = 0.134 |
| Phasic EMG × nadir epiglottic pressure | beta = 0.013, 95% CI [0.002, 0.024], $t$(149) = 2.296, $P$ = 0.002 (**) |
| Tonic EMG × nadir epiglottic pressure | beta = −0.040, 95% CI [−0.064, −0.017], $t$(149) = −3.366, $P$ < 0.001 (***) |

Dependent variable: regional tongue movement (mm).
The model accounted for repeated measures in participants (random effect) and adjusted for nadir epiglottic pressure. *$P$ <0.05, **$P$ < 0.01, ***$P$ < 0.001.

interactions (i.e. phasic EMG × nadir epiglottic pressure and tonic EMG × nadir epiglottic pressure, Table 4).

Furthermore, four groups were identified when categories of tongue dilatory movement and activity were considered together: (1) 'low' activity with minimal anterior movement (>1 mm) (101/156, 65%); (2) 'high' activity with large anterior movement (<1 mm) (5/156, 3%); (3) 'low' activity with large anterior movement (15/156, 10%); and (4) 'high' activity with minimal anterior movement (35/156, 22%).

Across groups 1 and 2, and while adjusting for nadir epiglottic pressure, a larger anterior movement was associated with a higher phasic EMG (i.e. EMG↗/mvt↗, Fig. 4A and Table 5). In contrast, for groups 3 and 4, a larger dilatory movement was associated with a lower phasic EMG (i.e. EMG↘/mvt↗, Fig. 4B and Table 5).

### Inspiratory genioglossus EMG and dilatory movement patterns

Twenty-one participants (21/46, 46%) had only EMG↗/mvt↗ compartments. Twenty-five participants (25/46, 54%) had at least one 'decoupled' EMG↘/mvt↗ compartment, including 17 with one, five with two, two with three and one with four EMG↘/mvt↗ compartments. Participant characteristics did not differ between participants with only EMG↗/mvt↗ compartments and those with at least one 'decoupled' EMG↘/mvt↗ (Mann–Whitney U-test, AHI $P$ = 0.161, BMI $P$ = 0.243, age $P$ = 0.987, tongue volume $P$ = 0.758, pharyngeal volume $P$ = 0.481, nasopharyngeal cross-sectional area $P$ = 0.155, soft palate length $P$ = 0.996, nadir epiglottic pressure $P$ = 0.457; Fisher's

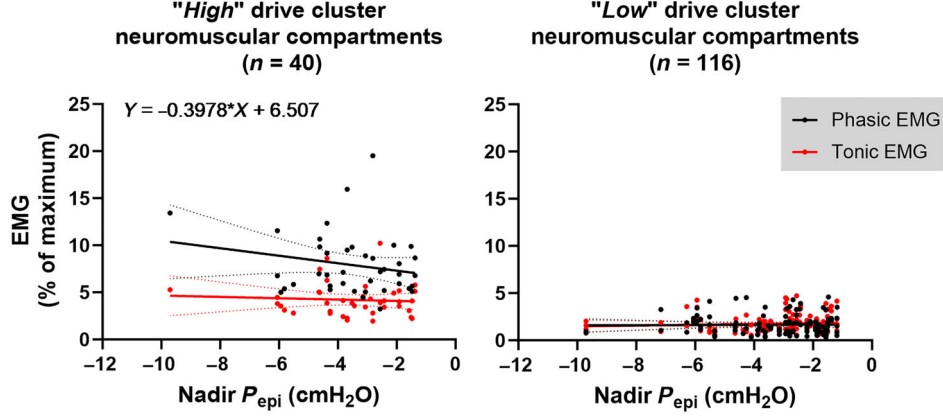

**Figure 3. In relationship between nadir epiglottic pressure ($P_{epi}$) and inspiratory genioglossus EMG for 'high' and 'low' activity clusters**
Among the 40 wires in the 'high' activity cluster, a larger negative nadir epiglottic pressure was associated with an increase in phasic EMG [mixed linear regression, beta = −1.711, 95% confident intervals [−2.792, −0.631], $t$(38) = −3.207, $P$ = 0.003]. This was not seen for the tongue neuromuscular compartments identified as having 'low' nor for tonic EMG in both activity clusters. Mixed linear regression analyses were used to assess associations and accounted for the fact that up to four measurements were made from each participant. Note: genioglossus EMG 'high' and 'low' activity clusters are described in Table 3. The simple linear regression with 95% confident intervals are plotted for each relationship and the equation reported only for the statistically significant relationship. [Colour figure can be viewed at wileyonlinelibrary.com]

exact test, gender proportion $P = 0.188$). However, when adjusting for nadir epiglottic pressure, OSA participants had a higher likelihood of having at least one decoupled EMG↘/mvt↗ compartment [binary logistic regression, odds ratio [95% CI]: 7.53 [1.19–47.47] ($P = 0.032$)] (Fig. 5). Only OSA participants had two or more EMG↘/mvt↗ compartments. The proportion of EMG↘/mvt↗ compartments did not differ between tongue neuromuscular compartments (Fisher's exact test, $P = 0.552$, horizontal posterior: 18/52, 34.6%, horizontal anterior: 8/34, 23.5%, oblique posterior: 7/25, 28.0%, and oblique anterior: 17/45, 37.8%). There was no association

between 'decoupling' status in a particular compartment and OSA status (Table 6).

Minimal movement patterns were seen in 83% of people without any EMG↘/mvt↗ compartments and were less common when one EMG↘/mvt↗ compartment was greater (65% with one and 50% with two or more EMG↘/mvt↗ compartments, Fig. 6). EMG↘/mvt↗ compartments were more common when the movement pattern was not minimal (i.e. oropharyngeal and en bloc movement patterns, see Methods). However, participants with at least one 'decoupled' EMG↘/mvt↗ compartment were not more likely to inspire with a not minimal

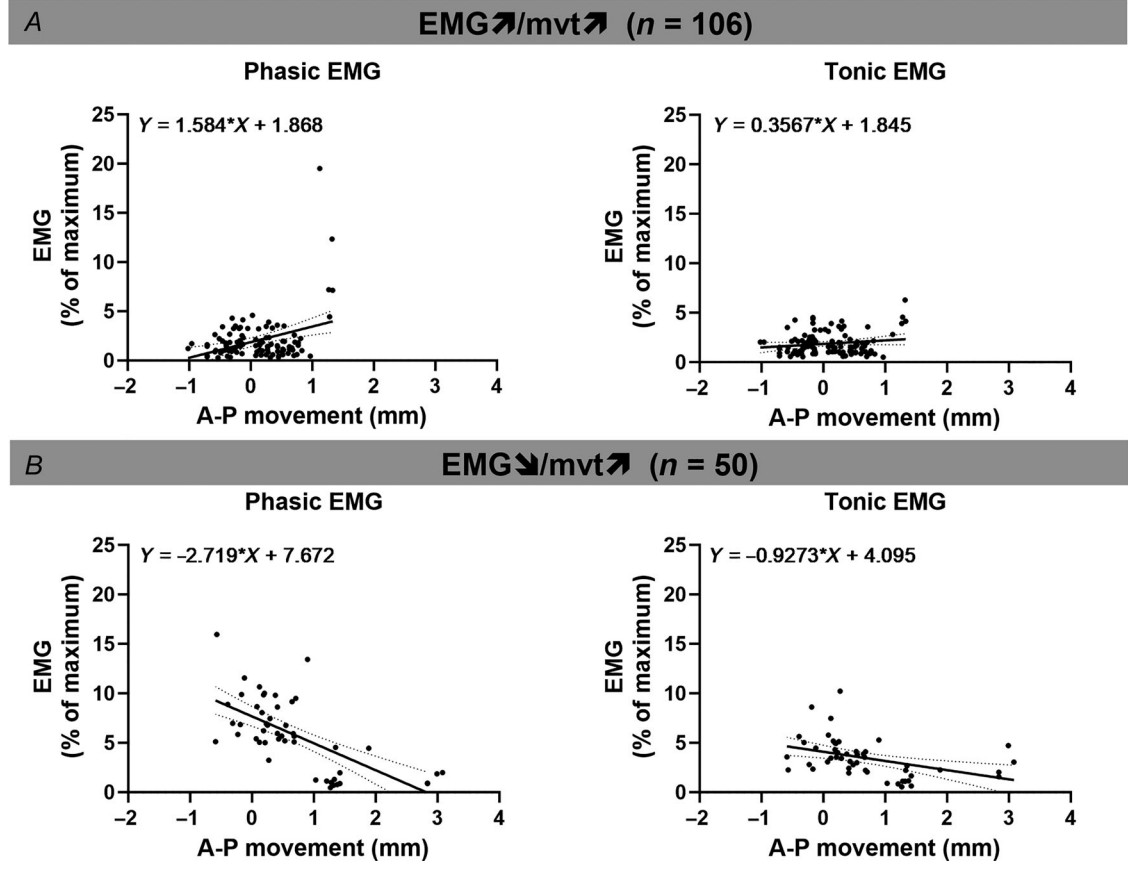

**Figure 4. Regional inspiratory tongue dilatory movement and EMG**

*A*, for 106 neuromuscular tongue compartments classified as having either a '*low*' activity with minimal anterior movement (>1 mm) or '*high*' activity with large anterior movement (<1 mm), a larger anterior (dilatory) tongue movement was associated with higher phasic EMG [beta = 0.089, 95% confidence intervals [0.000, 0.178], $t(99) = 1.995$, $P = 0.049$]. *B*, for the remaining compartments ($n = 50$) classified as having either a '*low*' activity with large anterior movement or a '*high*' activity with minimal anterior movement, a larger dilatory movement was associated with a lower phasic EMG [beta = −0.123, 95% confidence intervals [−0.224, −0.022], $t(43) = −2.458$, $P = 0.018$]. Tonic EMG was not related to anterior tongue movement in both cases. Mixed linear regressions accounting for the fact that up to four measurements were made from each participant (random effects) were used to assess associations between tongue dilatory movement (dependent variable) and EMG measurements and their interaction (fixed effects) with the covariate nadir epiglottic pressure (Table 5). Simple linear regression with 95% confident intervals are plotted and the equation reported on each graph. Note: positive displacement indicates anterior (dilatory) movement and '*low*' and '*high*' levels of activity are defined in Table 3.

**Table 5. Estimates of fixed effects of the mixed linear regression used to determine the effect of tonic and phasic EMG on inspiratory tongue movement for EMG↗/mvt↗ and EMG↘/mvt↗ tongue compartments**

| Parameter | Estimates of fixed effects for EMG↗/mvt↗ compartments (*n* = 106 compartments) |
|---|---|
| Intercept | beta = 0.156, 95% CI [−0.267, 0.579], *t*(99) = 0.731, *P* = 0.467 |
| Tonic EMG | beta = −0.064, 95% CI [−0.173, 0.044], *t*(99) = −1.172, *P* = 0.244 |
| Phasic EMG | beta = 0.089, 95% CI [0.000, 0.178], *t*(99) = 1.995, *P* = 0.049 (*) |
| Tonic × phasic EMG | beta = 0.001, 95% CI [−0.004, 0.006], *t*(99) = 0.200, *P* = 0.842 |
| Nadir epiglottic pressure | beta = 0.057, 95% CI [−0.059, 0.172], *t*(99) = 0.972, *P* = 0.333 |
| Phasic EMG × nadir epiglottic pressure | beta = 0.014, 95% CI [−0.006, 0.033], *t*(99) = 1.405, *P* = 0.163 |
| Tonic EMG × nadir epiglottic pressure | beta = −0.015, 95% CI [−0.045, 0.014], *t*(99) = −1.035, *P* = 0.303 |
| | Estimates of fixed effects for EMG↘/mvt↗ compartments (*n* = 50 compartments) |
| Intercept | beta = 1.482, 95% CI [0.444, 2.520], *t*(43) = 2.878, *P* = 0.006 |
| Tonic EMG | beta = −0.086, 95% CI [−0.260, 0.088], *t*(43) = −0.998, *P* = 0.324 |
| Phasic EMG | beta = −0.123, 95% CI [−0.224, −0.022], *t*(43) = −2.458, *P* = 0.018 (*) |
| Tonic × phasic EMG | beta = 0.007, 95% CI [−0.002, 0.017], *t*(43) = 1.533, *P* = 0.132 |
| Nadir epiglottic pressure | beta = −0.033, 95% CI [−0.300, 0.234], *t*(43) = −0.247, *P* = 0.806 |
| Phasic EMG × nadir epiglottic pressure | beta = 0.004, 95% CI [−0.011, 0.019], *t*(43) = 0.530, *P* = 0.599 |
| Tonic EMG × nadir epiglottic pressure | beta = −0.018, 95% CI [−0.058, 0.022], *t*(43) = −0.894, *P* = 0.376 |

Dependent variable: regional tongue movement (mm).
The model accounted for repeated measures in participants (random effect) and the covariate nadir epiglottic pressure. *$P < 0.05$.

movement pattern [binary logistic regression, odds ratio [95% CI]: 1.47 [0.95–19.91] ($P = 0.059$)].

## Discussion

### Main findings

This is the first study to assess directly the relationship between regional EMG to the tongue muscle and airway dilatation during awake quiet inspiration. It showed that there are complex and, in some cases, decoupled relationships between genioglossus activity and dilatory tongue movement. This finding suggests that upper airway dilatory mechanics cannot be predicted from genioglossus EMG as overall regional tongue movements do not directly correlate with genioglossus activity in all cases unless nadir epiglottic pressure is controlled for. The main results are: (1) a larger dilatory tongue

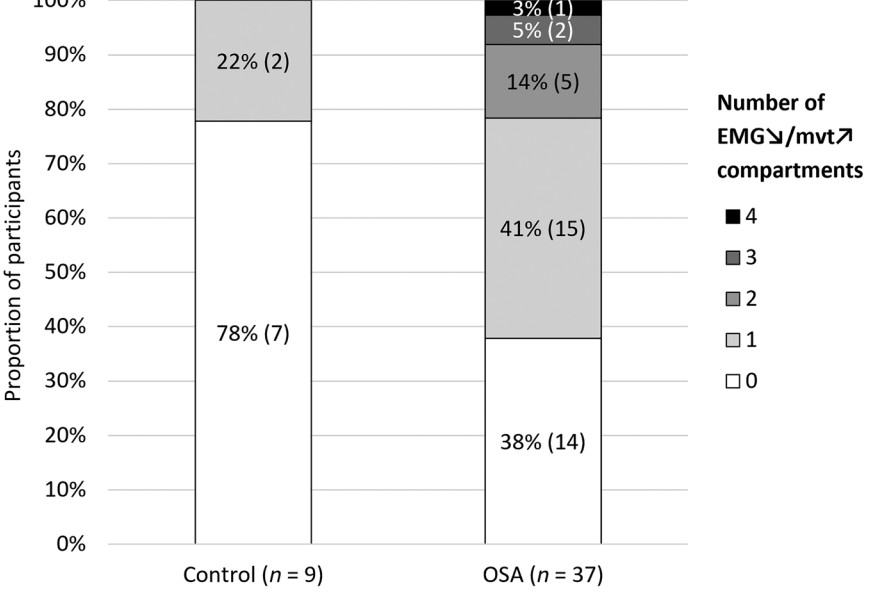

**Figure 5. Proportion of control and OSA participants with different numbers of decoupled EMG↘/mvt↗ neuromuscular compartments**
OSA participants had a higher likelihood of having at least one decoupled EMG↘/mvt↗ compartment [binary logistic regression, odds ratio [95% CI]: 7.53 [1.19–47.47] ($P = 0.032$)]. Data labels: % (*n* subjects).

**Table 6. Fixed effects of the generalised linear mixed model to determine the effect of OSA, tongue compartments and their interactions on the 'decoupling' status of the compartment**

| Parameter | 'Decoupling' status[†] |
|---|---|
| Corrected model | $F_{(8,147)} = 0.832$, $P = 0.576$ |
| OSA status | $F_{(1,50)} = 0.074$, $P = 0.786$ |
| Tongue compartment × OSA status | $F_{(3,147)} = 0.692$, $P = 0.558$ |
| Tongue compartment | $F_{(3,147)} = 0.013$, $P = 0.998$ |
| Nadir epiglottic pressure | $F_{(1,37)} = 1.761$, $P = 0.193$ |

[†]Dependent variable.
The model accounted for repeated measures in participants (random effect) and the covariate nadir epiglottic pressure (main effect).

movement was associated with a higher genioglossus phasic EMG for most (68%) of the genioglossus neuromuscular compartments (EMG↗/mvt↗) studied. Counterintuitively, a larger tongue dilatory movement was associated with a lower genioglossus phasic EMG (EMG↘/mvt↗) for the remaining compartments. (2) OSA participants had an ∼7-times higher likelihood of having at least one decoupled EMG↘/mvt↗ compartment. (3) When only EMG↗/mvt↗ neuromuscular compartments were observed in a participant, tongue dilatory movement was nearly always minimal. In contrast, large dilatory movements in the oropharyngeal and en bloc patterns occurred more in EMG↘/mvt↗ neuromuscular compartments. (4) In total, 26% of the neuromuscular compartments had a '*high*' level of

activity, and only for those was there a larger negative nadir epiglottic pressure associated with an increase in phasic EMG.

## Inspiratory genioglossus EMG and OSA pathophysiology

Spatial heterogeneity in genioglossus EMG during breathing has been previously documented in healthy subjects (Eastwood et al., 2003; Vranish & Bailey, 2015), and it has been assumed that different branches of the hypoglossal nerve (Mu & Sanders, 2010) could enable task-specific regional activation, as previously found by our group during tongue protrusion and swallowing (Yeung et al., 2022). Heterogeneous activation during breathing could therefore contribute to impaired muscle coordination and upper airway collapse during sleep. However, previous recordings of EMG bilaterally within the anterior and posterior sections of the genioglossus have conflicting results, with one study reporting more respiratory-related EMG anteriorly (Eastwood et al., 2003) and the other reporting more EMG posteriorly (Vranish & Bailey, 2015). In this study, we found no differences in inspiratory phasic and tonic EMG magnitude between compartments in people with or without OSA (Fig. 2 and Table 2). This finding suggests uniform inspiratory activity across the tongue during awake inspiration. Differences between studies might be due to recording location depth between studies since spatial differences in the inspiratory phasic and tonic motor units have been reported for superficial and deeper genioglossal recordings (Luu et al., 2018). In this study, the wire locations were chosen to target the anterior and posterior

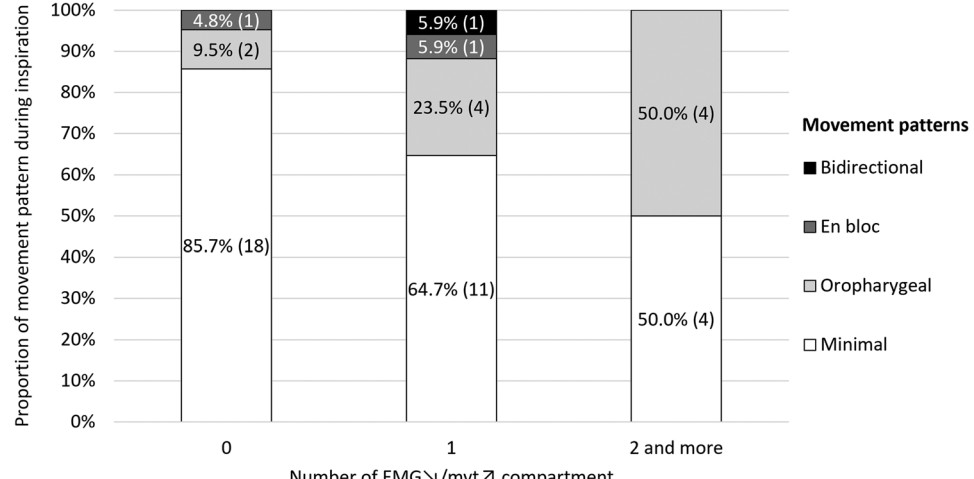

**Figure 6. Proportion of tongue movement patterns as a function of the number of EMG↘/mvt↗ tongue compartments**
Participants with at least one 'decoupled' EMG↘/mvt↗ compartment were not more likely to inspire with not minimal movement pattern (binary logistic regression, odds ratio [95% CI]: 1.47 [0.95–19.91] (*P* = 0.059). Data labels: % (*n* subjects).

sections of the oblique and horizontal compartments of the genioglossus close to the mid-sagittal plane to compare local activity with tongue movement in the same regions.

Two distinct levels of inspiratory genioglossus activity were observed, including a '*high*'-level phasic and tonic EMG during quiet nasal breathing in a subset of the compartments (26%). For comparison, in healthy adults, (Vranish & Bailey, 2015) reported multi-unit EMG of $\sim$ 6% of maximum, and Eastwood al. (2003) reported phasic and tonic EMG of $8.4 \pm 3.8\%$ and $4.5 \pm 1.4\%$ of maximum, respectively. These values are consistent with our '*high*' EMG level [phasic EMG: $7.91 \pm 3.27$, and tonic EMG: $4.20 \pm 1.75$ (% of maximum), Table 3]. It is unlikely that the '*low*' activity [phasic EMG: $1.64 \pm 1.06$, and tonic EMG: $1.76 \pm 1.00$ (% of maximum), Table 3] was the result of technical considerations such as a difference in orientation of the recording relative to the active muscle fibres as '*low*' activity was observed in similar proportions in all four compartments. It is also probably not due to our normalisation with the maximum EMG from either tongue protrusion or swallowing, as that did not differ between tongue compartments.

One possibility is that this '*high*' EMG is compensating for restricted anatomy (Mezzanotte et al., 1992), as compartments with '*high*' neuromuscular activity were more common in people with smaller airways (both volume and cross-sectional area). While a larger negative epiglottic pressure swing did not predict a high EMG overall, for those neuromuscular compartments in which there was unusually '*high*' EMG, the phasic EMG magnitude was associated with more negative epiglottic pressure (Fig. 3). Thus, in this case, it probably reflects an increase in the inspiratory drive to genioglossus via effective reflex facilitation evoked by negative pressure in the upper airway (Carberry et al., 2022; Horner et al., 1991; Wheatley et al., 1993). If this is confirmed, this may have implications for the identification of OSA patients with more effective reflex drive.

No difference in genioglossus activity between people with and without OSA was observed in this study, although there were relatively few control participants with sufficient data available to be included in the analysis. This contrasts with Mezzanotte et al. (1992) and Fogel et al. (2001), who reported greater multi-unit EMG in people with OSA compared to controls during wakefulness. The latter could be the result of a larger motor action potential area associated with neurogenic remodelling. Motor unit action potential size is increased in people with OSA, indicating mild neuropathy and higher numbers of muscle fibres in each motor unit (Saboisky et al., 2007, 2012, 2015), although genioglossus single motor unit firing rates do not differ between OSA and control participants in awake resting breathing seated and supine (Luu et al., 2020; Saboisky et al., 2007).

Mezzanotte et al. (1992) and Fogel et al. (2001) recruited OSA patients with an AHI of at least 25 events/hr and, on average, a higher BMI than those reported in this study ($41.6 \pm 2.3$ and $32.0 \pm 1.3$ kg/m$^2$, respectively, *vs.* $27.5 \pm 3.9$ kg/m$^2$ in the current study). This may have triggered a greater inspiratory multi-unit EMG in participants with OSA due to the higher degree of OSA severity and inspiratory loads (due to higher BMI) than in our OSA participants, which also includes participants with mild OSA. For comparison, our OSA participants exhibited similar level of phasic EMG to the healthy obese controls reported in Mezzanotte et al. (1992).

### Associations between regional inspiratory tongue dilatory movement and genioglossus EMG

Airway patency is preserved during wakefulness in people with OSA via active stiffening and dilatory movement of the tongue, with the magnitude and spatial extent of this motion being greater in people with more severe OSA (Juge, Knapman et al., 2020). However, how much the tongue can move to dilate the airway is influenced by both neural drive and local mechanics (Bilston & Gandevia, 2014). In this study, people with OSA had a larger tongue volume and more negative nadir epiglottic pressure than control participants, and the overall relationship between genioglossus EMG and anterior tongue dilatory movement suggests that people with less negative nadir epiglottic pressure move their tongue less during inspiration for a given activity (Table 4). A closer examination of the data also established that the genioglossus inspiratory activity and dilatory tongue movement relationship was decoupled (EMG$\searrow$/mvt$\nearrow$) in almost one-third (32%) of the neuromuscular compartments. OSA participants had an $\sim$7-times higher odds of having at least one decoupled EMG$\searrow$/mvt$\nearrow$ compartment than non-OS participants, which may contribute to the pathogenesis of OSA.

Change in tongue shape can be achieved by various different combinations of muscle contraction since it is a muscular hydrostat (Kier & Smith, 1985). In those tongue compartments with a decoupled EMG$\searrow$/mvt$\nearrow$ relationship, greater dilatory motion might occur without an increased regional genioglossus activity. A possible mechanism could be that the lower muscle activity makes that region more compliant, and thus more able to deform in response to activation of other tongue muscles. This could represent an alternative strategy to maintain patency in a particularly confined oral cavity. Tongue tissue has been reported to be softer in people with OSA (Brown et al., 2015) and might reflect a decrease in muscle tone in OSA. Alternatively, the capacity of genioglossus to generate force and dilatory movement may be impaired by changes in tongue muscle microstructure in people

with OSA related to inflammation and fibrosis (Boyd et al., 2004), atrophied muscle fibres (Friberg et al., 1998) or changes in muscle fibre types (Series et al., 1995), requiring co-activation of other muscles to maintain airway patency during inspiration (Cori et al., 2018). Intrinsic and extrinsic tongue muscles, including the genioglossus, the styloglossus, the hyoglossus, and intrinsic longitudinals, vertical and transverse muscles are potential candidates for such activity. This could explain, at least in part, how large dilatory movement of the tongue can be associated with minimal inspiratory phasic activity in the genioglossus. However, this needs to be verified, in particular during sleep. There is evidence that upper airway drive decreases for genioglossus and non-genioglossus airway dilator muscles at sleep onset, although the magnitude of the reduction differs between muscles and drive recovers after sleep onset in many cases (Cori et al., 2018).

Nearly all participants with only EMG↗/mvt↗ tongue compartments exhibited minimal dilatory movement (>1 mm in both oblique and horizontal compartments). We interpret this as an indication that phasic EMG in these participants acts to stabilise the tongue position and stiffen the tongue isometrically. This could be an effective strategy for people whose upper airway geometry is minimally compromised, and airflow is adequate without additional dilatation, although the participant characteristics did not differ between those with and without decoupled EMG↘/mvt↗ compartments in this study. Previous observations have reported that minimal tongue dilatory movement occurred in people with a large upper airway cross-sectional area, with or without OSA (Cheng et al., 2014; Juge, Knapman et al., 2020). However, for people with a narrow upper airway, minimal tongue movement could indicate anatomical tongue confinement in the oral cavity, preventing or limiting tongue dilatory movement with increased genioglossus activity (Bilston & Gandevia, 2014).

## Limitations

First, some EMG wires were not inserted in the intended neuromuscular compartment into the genioglossus despite standardised placement protocols, probably due to anatomical variation between individuals. However, those wires were reassigned to the appropriate neuromuscular compartment based on the EMG recordings during swallowing, which differentiates the oblique and horizontal compartments (Yeung et al., 2022). Second, phasic and tonic EMG were normalised to the highest peak EMG during either tongue protrusion or swallowing in each compartment, evoked with verbal encouragement, as this method produced consistent maximal EMG across all tongue regions. However, it cannot be guaranteed that these are truly 'maximal'. Third, EMG recordings for each neuromuscular compartment represent activity from a small muscle volume surrounding the tip of the wire, which may not represent the full array of activity within each neuromuscular compartment (Aldes, 1995). Fourth, genioglossus EMG and anterior inspiratory movement were not measured simultaneously since intramuscular wires could not be inserted in the muscle while the participants were in the scanner for both MR safety and image quality reasons. However, efforts were made to position participants for EMG assessment so that their head posture was similar to conditions in the MRI. In both cases, participants were also instructed to breathe quietly through their nose as the MR respiratory sensor used cannot quantify respiratory effort. Fifth, airway cross-sectional area and pharyngeal volume represent the average size across the whole respiratory cycle as they were measured on anatomical and mDIXON images, respectively, that are collected over multiple respiratory cycles. Finally, only genioglossus EMG was assessed. Thus, we could not establish how co-activation of other upper airway dilator muscles probably contributed to large anterior movement of the tongue with minimal EMG genioglossus activity.

## Conclusions

Relationships between genioglossus activity and tongue movement during wakefulness are complex and vary between individuals. We hypothesised that higher genioglossus EMG would be associated with larger dilatory tongue movement, and this was the case in approximately two-thirds (68%) of the tongue neuromuscular compartments (EMG↗/mvt↗). For the remaining third (32%) of compartments, a larger dilatory movement was associated with a lower phasic EMG. This may contribute to OSA pathogenesis, as OSA participants had an ∼7-times higher likelihood of having at least one compartment for which a larger anterior tongue movement was not associated with a higher phasic EMG than controls. This has important implications for upper airway control as it demonstrates that upper airway muscle movement cannot be directly inferred from EMG measurements alone and vice versa, particularly in people with OSA for whom larger anterior tongue dilatory movement can be achieved for a given EMG compared to controls with less negative nadir epiglottic pressure. Furthermore, the results also demonstrate that large anterior tongue dilatory movement can be associated with minimal local genioglossus inspiratory activity, suggesting significant co-activation of other dilator muscles to dilate the airway.

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

## Additional information

### Data availability statement

Data are available on request due to privacy/ethical restrictions.

## Competing interests

Outside the submitted work, D.J.E. has had research grants from Bayer, Takeda, Invicta Medical and Apnimed and has served on Scientific Advisory Boards for Apnimed, Invicta, Mosanna and as a consultant for Bayer. None of the other authors has any conflicts of interests.

## Author contributions

All authors have approved the final version of the manuscript and agree to be accountable for all aspects of the work. All persons designated as authors qualify for authorship, and all those who qualify for authorship are listed.

## Funding

This research was funded by the National Health & Medical Research Council (NHMRC) of Australia (#APP1058974). Lynne E. Bilston, S. C. Gandevia, D. J. Eckert and J. E. Butler are supported by NHMRC Fellowships (#APP1077934, APP1078061, APP1196261 and APP1042646, respectively).

## Acknowledgements

The authors thank the NeuRA imaging centre for their technical support and Dr Peter Humburg, a statistical consultant from the University of New South Wales Mark Wainwright Analytical Centre (Stats Central), for his support with the study's statistical analysis.

Open access publishing facilitated by University of New South Wales, as part of the Wiley - University of New South Wales agreement via the Council of Australian University Librarians.

## Keywords

intramuscular EMG, magnetic resonance imaging, respiratory physiology, sleep disordered breathing, upper airway

## Supporting information

Additional supporting information can be found online in the Supporting Information section at the end of the HTML view of the article. Supporting information files available:

**Peer Review History**

