## [Peer Review History · The Journal of Physiology]

Regional association between inspiratory tongue dilatory movement and genioglossus activity during wakefulness in people with obstructive sleep apnoea

Lauriane Jugé, Angela Liao, Jade Yeung, Fiona Knapman, Christopher Bull, Peter GR Burke, Elizabeth C. Brown, Simon C Gandevia, Danny J Eckert, Jane E Butler, and Lynne E Bilston

DOI: 10.1113/JP285187

Corresponding author(s): Lynne Bilston (l.bilston@neura.edu.au)

The following individual(s) involved in review of this submission have agreed to reveal their identity: Ken D O'Halloran (Referee #1)

Review Timeline:

Submission Date:	21-Jun-2023
Editorial Decision:	14-Aug-2023
Revision Received:	25-Sep-2023
Accepted:	20-Oct-2023

Senior Editor: Scott Powers

Reviewing Editor: Frank Powell

Transaction Report:

Dear Professor Bilston,

Re: JP-RP-2023-285187 "Regional associations between inspiratory tongue dilatory movement and genioglossus drive during wakefulness in people with obstructive sleep apnoea" by Lauriane Jugé, Angela Liao, Jade Yeung, Fiona Knapman, Christopher Bull, Peter GR Burke, Elizabeth C. Brown, Simon C Gandevia, Danny J Eckert, Jane E Butler, and Lynne E Bilston

Thank you for submitting your manuscript to The Journal of Physiology. It has been assessed by a Reviewing Editor and by 2 expert referees and we are pleased to tell you that it is acceptable for publication following satisfactory revision.

LANGUAGE EDITING AND SUPPORT FOR PUBLICATION: If you would like help with English language editing, or other article preparation support, Wiley Editing Services offers expert help, including English Language Editing, as well as translation, manuscript formatting, and figure formatting at www.wileyauthors.com/eoo/preparation. You can also find resources for Preparing Your Article for general guidance about writing and preparing your manuscript at www.wileyauthors.com/eoo/prepresources.

REVISION CHECKLIST:

Please upload two versions of your manuscript text: one with all relevant changes highlighted and one clean version with no changes tracked. The manuscript file should include all tables and figure legends, but each figure/graph should be uploaded as separate, high-resolution files. The journal is now integrated with Wiley's Image Checking service. For further details, see: <https://www.wiley.com/en-us/network/publishing/research-publishing/trending-stories/upholding-image-integrity-wileys-image-screening-service>.

We look forward to receiving your revised submission.

Yours sincerely,

Scott K. Powers
Senior Editor
The Journal of Physiology
<https://jp.msubmit.net>
<http://jp.physoc.org>
The Physiological Society
Hodgkin Huxley House
30 Farringdon Lane
London, EC1R 3AW
UK
<http://www.physoc.org>
<http://journals.physoc.org>

REQUIRED ITEMS

- Author photo and profile. First (or joint first) authors are asked to provide a short biography (no more than 100 words for one author or 150 words in total for joint first authors) and a portrait photograph. These should be uploaded and clearly labelled with the revised version of the manuscript. See Information for Authors for further details.
- The contact information provided for the person responsible for 'Research Governance' at your institution is an author on this paper. Please provide an alternative contact who is not an author on this paper or confirm that the author whose email was provided has sole responsibility for research governance. This is the person who is responsible for regulations, principles and standards of good practice in research carried out at the institution, for instance the ethical treatment of animals, the keeping of proper experimental records or the reporting of results.
- The Journal of Physiology funds authors of provisionally accepted papers to use the premium BioRender site to create high resolution schematic figures. Follow this link and enter your details and the manuscript number to create and download figures. Upload these as the figure files for your revised submission. If you choose not to take up this offer we require figures to be of similar quality and resolution. If you are opting out of this service to authors, state this in the Comments section on the Detailed Information page of the submission form. The link provided should only be used for the purposes of this submission. Authors will be charged for figures created on this premium BioRender account if they are not related to this manuscript submission.
- Please upload separate high-quality figure files via the submission form.
- Please ensure that any tables are in Word format and are, wherever possible, embedded in the article file itself.
- Please ensure that the Article File you upload is a Word file.
- Papers must comply with the Statistics Policy: https://jp.msubmit.net/cgi-bin/main.plex?form_type=display_requirements#statistics.

In summary:

- If n {less than or equal to} 30, all data points must be plotted in the figure in a way that reveals their range and distribution. A bar graph with data points overlaid, a box and whisker plot or a violin plot (preferably with data points included) are acceptable formats.
- If $n > 30$, then the entire raw dataset must be made available either as supporting information, or hosted on a not-for-profit repository e.g. FigShare, with access details provided in the manuscript.
- 'n' clearly defined (e.g. x cells from y slices in z animals) in the Methods. Authors should be mindful of pseudoreplication.

- All relevant 'n' values must be clearly stated in the main text, figures and tables.
- The most appropriate summary statistic (e.g. mean or median and standard deviation) must be used. Standard Error of the Mean (SEM) alone is not permitted, unless justified and presented alongside confidence intervals.
- Exact p values must be stated. Authors must not use 'greater than' or 'less than'. Exact p values must be stated to three significant figures even when 'no statistical significance' is claimed.
- Please include an Abstract Figure file, as well as the figure legend text within the main article file. The Abstract Figure is a piece of artwork designed to give readers an immediate understanding of the research and should summarise the main conclusions. If possible, the image should be easily 'readable' from left to right or top to bottom. It should show the physiological relevance of the manuscript so readers can assess the importance and content of its findings. Abstract Figures should not merely recapitulate other figures in the manuscript. Please try to keep the diagram as simple as possible and without superfluous information that may distract from the main conclusion(s). Abstract Figures must be provided by authors no later than the revised manuscript stage and should be uploaded as a separate file during online submission labelled as File Type 'Abstract Figure'. Please ensure that you include the figure legend in the main article file. All Abstract Figures should be created using BioRender. Authors should use The Journal's premium BioRender account to export high-resolution images. Details on how to use and access the premium account are included as part of this email.

EDITOR COMMENTS

Reviewing Editor:

Your submission has been reviewed by two experts in the field and both find considerable merit in the study, which is well-done. Congratulations. However, they also make several comments and raise issues that could be addressed in a revision to improve the manuscript.

Also, please double check that your results and figures comply with the journal requirements (raw data for example in Fig. 2?).

Methods section note - Please see some minor comments by the expert referees.

Senior Editor:

Thank you for submitting your work to the Journal of Physiology. Your report has been carefully reviewed by two referees and a review editor. All parties provide positive comments about your work but each of the referees have offered several suggestions for improvement of your work. We look forward to receiving your revised manuscript.

Statistics policy note - One figure (histogram) does not contain original data points.

REFEREE COMMENTS

Referee #1:

This is a very interesting study exploring the relationship between genioglossus motor drive (inferred from intramuscular EMG activity normalised to presumed maximum activity during swallowing) and dilatory movement (based on MRI) in regional compartments of the tongue during wakefulness in control participants and men and women with mild-to-severe obstructive sleep apnoea. I commend the authors on completing a technically challenging study, which notwithstanding limitations acknowledged by the authors, provides a considerable advance in our understanding of the complex neuromuscular system regulating tongue movement, relevant to various behaviours, including control of airway patency.

The manuscript is well written and presented. The results are clear and conclusions drawn on the basis of the findings are fair. The Discussion is balanced. Whereas, the marriage (coupling) of genioglossus motor activity and dilatory movement is relatively strong, a larger than expected divorce rate (uncoupling) was uncovered in one third of all tongue compartments analysed, characterised as large dilatory movements despite (or perhaps related to!) low genioglossus EMG activity. Uncoupling was more common in OSA, with more than half of people with OSA (62%) showing uncoupling in at least one tongue compartment (compared with 22% of controls). The capacity for large anterior dilatory movements associated with low genioglossus EMG activity reveals that the co-activation of muscles working in concert with the genioglossus is an important determinant of airway dilation, complicated further in the context of airway defence by the need to also consider the critical aspect of airway stiffness.

Was there a relationship between the extent of decoupling in tongue compartments and severity of OSA? Or was there an association between decoupling in a particular compartment and OSA severity, noting that the authors have previously reported (Juge et al. 2020) that genioglossus movement is more complex in OSA.

Since contemporaneous measurements were not performed, for technical reasons explained by the authors, I suggest you avoid statements that rather temptingly couple the observations such as in the conclusion where it is stated that "large...movement...in the presence of minimal local....drive...".

Measurements were taken during quiet breathing in wakefulness. Whereas an obvious extension of the work is characterisation of the relationship during sleep and/or changes in response to pharmacological intervention, where one might delineate contributions to airway collapse, I wonder, from the data in hand, if augmented breaths could be analysed to delineate neuromuscular and movement coupling/uncoupling during a presumed increase in airway calibre that might help extend understanding of the complex interactions. Spontaneous augmented breaths should be easily identifiable from recordings, particularly if the respiratory sensor used allows for a surrogate assessment of tidal volume or similar. Did the study design include any respiratory challenges? A sniff test or maximum inspiratory effort would have been interesting.

Minor

Pg 7 end of para 1: What sensor was used?

Pg 9: Were swallows 'dry' or induced by a liquid bolus?

Referee #2:

Jugé et al present a novel study describing regional genioglossus movement and neural activity levels in OSA patients, providing unique insight into reduced neuromuscular efficiency (failure to translate genioglossus activity into improved mechanics) that is a cause of OSA in many individuals. The study incorporates a rare combination of functional imaging and intramuscular EMG with pharyngeal pressure measurements. These physiological data are unique and highly valuable for our field, particularly given the highly ambitious and nature of the data collection. Investigators demonstrated commitment to rigor, with a notable example being the rescoring of sleep study data from outside their sites. Below I have detailed concerns that I feel could be addressed to even further raise the quality of the work. There are several ways modern statistical methods could be employed that I believe would have a notable impact on the findings.

1) The study is a descriptive physiological study that describes regional heterogeneity in phasic genioglossus activation and accompanying inspiratory movement (or the surprising lack therefore) across compartments. The first key message is that genioglossus activity (EMG) does not consistently translate into forward movement, even during wakefulness. I believe this message should come first in the abstract, and be supported by quantitative results describing how greater GG activity levels were not associated with increased movement levels. Currently the authors have used partial correlation analysis in Results, but ideally this analysis would involve a linear mixed model analysis describing anterior movement as a function of genioglossus activity (fixed effect) and subject (random effect) to take proper advantage of the repeated measurement design. The authors will still be expected to find that there is no strong association between movement and genioglossus activity in general (see also 6 below for adjustments for additional confounders).

2) To me, the second key physiological insight is that there is an association between forward genioglossus movement and EMG in some compartments but not others, which is a novel and important message.

a. First, the analysis needed for the definition of "EMG/mvt" is overly abbreviated in Methods and needs clarification. This subgroup of "EMG/mvt" appears to be defined based on "groups 1 and 2" in Results but should be defined more precisely in Methods and again in the appropriate Figure legend for clarity. Otherwise reader may interpret these incorrectly as (1) a simple anterior dilatory movement (since phasic EMG is always positive), or (2) a positive within-breath time-series correlation between EMG and movement synchronized with the respiratory cycle (note that the term "correlation" is used in the definition but no strict correlation is performed for the definition). Please also replace EMG/mvt with EMG/mvt; presumably there is no drop in EMG per se (i.e. negative phasic activity), rather the movement is considered the dependent variable in this analysis. Instead, of EMG/mvt the authors could consider using "efficient"/"inefficient" as short-hand for the reduced neuromuscular efficiency these compartments can be concluded to exhibit.

b. Re: "For 106 of the compartments (68%), a larger anterior (dilatory) movement was associated with a higher phasic EMG ($r = -0.26$, $P = 0.01$, hereafter EMG/mvt). In contrast, for the remaining 50 (32%) compartments, a larger dilatory movement was associated with a lower phasic EMG ($r = 0.49$, $P < 0.001$, hereafter EMG/mvt)." As these groups are defined by their EMG/mvt status, the direction of the correlation appears to be 'by definition' positive and negative in magnitude; this means that the main quantitative parameter of interest is how many compartments are in each group. I believe it is fine to report the R-values (and 95CI in R if desired) but I would remove this from the Abstract, and certainly remove p-values throughout as the p-values do not capture how these groups were selected using EMG and directional movement behavior (that are then used within the correlation). I don't believe this hurts the work in any way.

3) Perhaps the most important key message made by the authors is that, compared to controls, OSA patients appear to have a higher proportion of compartments with inefficient (EMG/mvt) translation from GG to anterior movement.

However, the analysis performed is not sufficiently powerful. With the data available in Figure 5, a preliminary analysis using multinomial ordinal logistic regression revealed that there is a meaningful association between OSA status and a higher likelihood of an increased number of inefficient (EMG/mvt) compartments, odds ratio [95%CI]: 6.2 [1.12-35] (p=0.037). The authors should perform a similar analysis in place of the Fisher Exact test, and consider including the finding in the abstract.

4) As noted by the authors, the two-way ANOVA analysis of whether compartment type or OSA status explains genioglossus activity (Figure 2) does not take advantage of the repeated measures across subjects, and also lacks power and flexibility. Linear mixed effects models analysis would be more appropriate here and would account for the duplicate measures within a compartment and the missing compartment data. There appears to be a signal for increased activity and posterior compartments that would be of interest. (First, model GG as a function of compartment type as a fixed effect, subject as a random effect; Second, add OSA status to the first model; Third add additional adjustment for confounders, see 6 below)

5) "Upper airway function" is used in place of "upper airway movement" throughout the manuscript; immediate concern is that the movement is being measured in most analyses herein. The authors will be aware that absence of movement may not imply failure of function depending on the pressures/forces that the region of interest is being exposed to. e.g. higher EMG could fail to generate anterior movement but be successfully increasing forces and preventing posterior movement in the presence of exposure to larger negative pressures; function might be normal but movement would be absent. Can the authors be more judicious with their use of "function" versus "movement" in interpretation where the totality of their data supports reduced anterior movement alone or when unadjusted for pressures.

6) To provide further data supporting use of 'function' moreso that 'movement' alone: While it is clear that OSA patients have a higher likelihood of inefficient GG compartments, it is not entirely clear that the failure to translate GG into movement can not be at least in part attributable to greater baseline anatomical compromise; adjusting analysis for Pepi swing magnitude and end-expiratory cross sectional area would help to address this and enable authors to more strongly make conclusions based on function. I strongly recommend adding these variables in sensitivity analysis as confounders to the mixed model and ordinal regression analyses recommended above. For example, it is plausible that the analysis recommended in (1) above would demonstrate that, after adjusting for these confounders, greater GG activity is associated with greater anterior (i.e. less posterior) movement. It is also plausible that the analysis recommended in (3) above would demonstrate an attenuation of the odds for greater number of inefficient compartments when these confounders are accounted for. If so we might learn that OSA is associated with greater inefficiency per se, or is associated with greater inefficiency in part via a greater baseline anatomical compromise. Both findings are of interest.

Minor

Figure 2. Can the y-axes be made to have the same magnitude for consistency? Tonic EMG has a larger y-range.

Is it confusing to have the "A-P movement" variable quantitatively described by negative values for the anterior direction. Can you rename to "horizontal posterior movement" or similar so that the defined direction is not the exact reverse of the intuitive direction? Even so, it will still remain somewhat challenging that the "mvt" describes increased anterior movement but values shown are negative. Perhaps just make the anterior values positive to solve all the issues here?

I believe Mathworks is in Natick rather than Danvers.

Prefer the use of genioglossus "activity" rather than genioglossus "drive"

Re: "However, it is unclear why some are unable to activate their tongue muscles to maintain airway patency during sleep (Younes, 2008)." Consider also citing Gell et al 2022 Thorax who provided some supplemental data on patients with neuromechanical inefficiency if you think it is appropriate.

Figure 3 legend. Accounting-> accounted (switch to consistent past tense).

AHI definition was not provided; can the authors comment on the criteria (AHI3pa vs AHI4)?

END OF COMMENTS

Confidential Review

21-Jun-2023

JP-RP-2023-285187

"Regional associations between inspiratory tongue dilatory movement and genioglossus drive during wakefulness in people with obstructive sleep apnoea"

by Lauriane Jugé, Angela Liao, Jade Yeung, Fiona Knapman, Christopher Bull, Peter GR Burke, Elizabeth C. Brown, Simon C Gandevia, Danny J Eckert, Jane E Butler, and Lynne E Bilston

REQUIRED ITEMS

- Author photo and profile. First (or joint first) authors are asked to provide a short biography (no more than 100 words for one author or 150 words in total for joint first authors) and a portrait photograph. These should be uploaded and clearly labelled with the revised version of the manuscript. See Information for Authors for further details.

Dr. Lauriane Jugé and Ms Angela Lio (co-first author) have provided a short biography and portrait photos.

- The contact information provided for the person responsible for 'Research Governance' at your institution is an author on this paper. Please provide an alternative contact who is not an author on this paper or confirm that the author whose email was provided has sole responsibility for research governance. This is the person who is responsible for regulations, principles and standards of good practice in research carried out at the institution, for instance the ethical treatment of animals, the keeping of proper experimental records or the reporting of results.

Mrs Deborah McKay is the Research Governance and Compliance Manager at NeuRA. She is the appropriate contact and not an author of this paper.

Neuroscience Research Australia (NeuRA)
Margarete Ainsworth Building
Barker Street Randwick Sydney NSW 2031 Australia
T +61 2 9399 1676
d.mckay@neura.edu.au

- The Journal of Physiology funds authors of provisionally accepted papers to use the premium BioRender site to create high resolution schematic figures. Follow this link and enter your details and the manuscript number to create and download figures. Upload these as the figure files for your revised submission. If you choose not to take up this offer we require figures to be of similar quality and resolution. If you are opting out of this service to authors, state this in the Comments section on the Detailed Information page of the submission form. The link provided should only be used for the purposes of this submission. Authors will be charged for figures created on this premium BioRender account if they are not related to this manuscript submission.

BioRender was used to prepare a high-quality abstract figure.

- Please upload separate high-quality figure files via the submission form.

High-resolution Figures 1 to 6 versions have been uploaded and prepared with a width of either 17.5 or 8.5 cm. Figures captions are listed at the end of the article file.

- Please ensure that any tables are in Word format and are, wherever possible, embedded in the article file itself. Please ensure that the Article File you upload is a Word file.

Six tables are embedded in the manuscript (word format).

- Papers must comply with the Statistics Policy: https://jp.msubmit.net/cgi-bin/main.plex?form_type=display_requirements#statistics.

In summary:

- If n {less than or equal to} 30, all data points must be plotted in the figure in a way that reveals their range and distribution. A bar graph with data points overlaid, a box and whisker plot or a violin plot (preferably with data points included) are acceptable formats.
- If $n > 30$, then the entire raw dataset must be made available either as supporting information, or hosted on a not-for-profit repository e.g. FigShare, with access details provided in the manuscript.
- 'n' clearly defined (e.g. x cells from y slices in z animals) in the Methods. Authors should be mindful of pseudoreplication.
- All relevant 'n' values must be clearly stated in the main text, figures and tables.
- The most appropriate summary statistic (e.g. mean or median and standard deviation) must be used. Standard Error of the Mean (SEM) alone is not permitted, unless justified and presented alongside confidence intervals.
- Exact p values must be stated. Authors must not use 'greater than' or 'less than'. Exact p values must be stated to three significant figures even when 'no statistical significance' is claimed.

The manuscript complies with the statistics policy. All data points are plotted on the figures. "n" are clearly defined. Results are reported as mean \pm standard deviation [minimum-maximum]. P-values have been rectified to show three significant figures even when 'no statistical significance' is being reported.

- Please include an Abstract Figure file, as well as the figure legend text within the main article file. The Abstract Figure is a piece of artwork designed to give readers an immediate understanding of the research and should summarise the main conclusions. If possible, the image should be easily 'readable' from left to right or top to bottom. It should show the physiological relevance of the manuscript so readers can assess the importance and content of its findings. Abstract Figures should not merely recapitulate other figures in the manuscript. Please try to keep the diagram as simple as possible and without superfluous information that may distract from the main conclusion(s). Abstract Figures must be provided by authors no later than the revised manuscript stage and should be uploaded as a separate file during online submission labelled as File Type 'Abstract Figure'. Please ensure that you include the figure legend in the main article file. All Abstract Figures should be created using BioRender. Authors should use The Journal's premium BioRender account to export high-resolution images. Details on how to use and access the premium account are included as part of this email.

An abstract figure has been attached to the revised manuscript.

EDITOR COMMENTS

Reviewing Editor:

Your submission has been reviewed by two experts in the field and both find considerable merit in the study, which is well-done. Congratulations. However, they also make several comments and raise issues that could be addressed in a revision to improve the manuscript.

The team warmly thanks the reviewers and the editors for all their positive feedback, constructive comments, and suggestions to improve the clarity and the quality of this manuscript. We have addressed each point raised, and the manuscript has been revised, as described below.

Also, please double check that your results and figures comply with the journal requirements (raw data for example in Fig. 2?).

The Figure 2 has been amended to show the data points overlaid with mean \pm standard deviation.

Methods section note - Please see some minor comments by the expert referees.

Senior Editor:

Thank you for submitting your work to the Journal of Physiology. Your report has been carefully reviewed by two referees and a review editor. All parties provide positive comments about your work but each of the referees have offered several suggestions for improvement of your work. We look forward to receiving your revised manuscript.

Thank you.

Statistics policy note - One figure (histogram) does not contain original data points.

Figure 2 has been amended to comply with the statistics policy. It now shows the individual data points along with the mean \pm standard deviation. P-values have also been rectified to show 3 figures even when not statistically significant.

REFEREE COMMENTS

Referee #1:

This is a very interesting study exploring the relationship between genioglossus motor drive (inferred from intramuscular EMG activity normalised to presumed maximum activity during swallowing) and dilatory movement (based on MRI) in regional compartments of the tongue during wakefulness in control participants and men and women with mild-to-severe obstructive sleep apnoea. I commend the authors on completing a technically challenging study, which notwithstanding limitations

acknowledged by the authors, provides a considerable advance in our understanding of the complex neuromuscular system regulating tongue movement, relevant to various behaviours, including control of airway patency.

The manuscript is well written and presented. The results are clear and conclusions drawn on the basis of the findings are fair. The Discussion is balanced. Whereas, the marriage (coupling) of genioglossus motor activity and dilatory movement is relatively strong, a larger than expected divorce rate (uncoupling) was uncovered in one third of all tongue compartments analysed, characterised as large dilatory movements despite (or perhaps related to!) low genioglossus EMG activity. Uncoupling was more common in OSA, with more than half of people with OSA (62%) showing uncoupling in at least one tongue compartment (compared with 22% of controls). The capacity for large anterior dilatory movements associated with low genioglossus EMG activity reveals that the co-activation of muscles working in concert with the genioglossus is an important determinant of airway dilation, complicated further in the context of airway defence by the need to also consider the critical aspect of airway stiffness.

Was there a relationship between the extent of decoupling in tongue compartments and severity of OSA? Or was there an association between decoupling in a particular compartment and OSA severity, noting that the authors have previously reported (Juge et al. 2020) that genioglossus movement is more complex in OSA.

The study is not sufficiently powered to determine whether increased OSA severity was associated with a higher number of “decoupled” compartments. However, in the revised manuscript we have now indicated on page 17 that when adjusting for nadir epiglottic pressure, OSA participants had a higher likelihood of having at least one decoupled EMG↘/mvt↗ compartment (binary logistic regression, odds ratio [95%CI]: 7.53 [1.19 - 47.47] (P = 0.032)).

We also now report on page 17 that:

- 1- The proportion of EMG↘/mvt↗ compartments did not differ between tongue neuromuscular compartments (Fisher’s exact test, P = 0.552, horizontal posterior: 18/52, 34.6%, horizontal anterior: 8/34, 23.5%, oblique posterior: 7/25, 28.0%, and oblique anterior: 17/45, 37.8%).
- 2- There was no association between the ‘decoupling’ status in a particular compartment and OSA status using a generalised linear mixed model. Please see also Table 6 in the manuscript on page 17, and SPSS tables output below for your information.

Dependent variable: “decoupling status”, i.e. whether the compartment has been classified as EMG↗/mvt↗ or EMG↘/mvt↗.

Fixed Effects ^a				
Source	F	df1	df2	Sig.
Corrected Model	.832	8	147	.576
OSA_status	.074	1	50	.786
OSA_status * tongue_compartment	.692	3	147	.558
tongue_compartment	.013	3	147	.998
P_nadir	1.761	1	37	.193

Probability distribution: Binomial

Link function: Logit

a. Target: dis_orientation

Pairwise Contrasts

tongue_compartment	OSA_status Pairwise Contrasts	Contrast Estimate	Std. Error	t	df	Adj. Sig.	95% Confidence Interval	
							Lower	Upper
GGo anterior	Control - OSA	.196	.186	1.054	131	.294	-.171	.563
	OSA - Control	-.196	.186	-1.054	131	.294	-.563	.171
GGo posterior	Control - OSA	-.112	.257	-.437	141	.662	-.620	.396
	OSA - Control	.112	.257	.437	141	.662	-.396	.620
GGh anterior	Control - OSA	-.254	.346	-.735	147	.464	-.937	.429
	OSA - Control	.254	.346	.735	147	.464	-.429	.937
GGh posterior	Control - OSA	.025	.220	.113	147	.910	-.409	.459
	OSA - Control	-.025	.220	-.113	147	.910	-.459	.409

The sequential Sidak adjusted significance level is .05.
Confidence interval bounds are approximate.

Since contemporaneous measurements were not performed, for technical reasons explained by the authors, I suggest you avoid statements that rather temptingly couple the observations such as in the conclusion where it is stated that "large...movement...in the presence of minimal local....drive...".

Fair point. The wording has been changed throughout the manuscript where needed.

E.g. in the abstract: " Tongue movement associated with minimal genioglossus activity suggests co-activation of other airway dilator muscles".

Measurements were taken during quiet breathing in wakefulness. Whereas an obvious extension of the work is characterisation of the relationship during sleep and/or changes in response to pharmacological intervention, where one might delineate contributions to airway collapse, I wonder, from the data in hand, if augmented breaths could be analysed to delineate neuromuscular and movement coupling/uncoupling during a presumed increase in airway calibre that might help extend understanding of the complex interactions. Spontaneous augmented breaths should be easily identifiable from recordings, particularly if the respiratory sensor used allows for a surrogate assessment of tidal volume or similar. Did the study design include any respiratory challenges? A sniff test or maximum inspiratory effort would have been interesting.

Indeed, there are exciting potential future work as the results of this study that can be conducted, including also better understand co-activation of upper airway dilators.

Yes, this is right. Unfortunately, no respiratory challenges were included in the protocol, and our respiratory sensor does not permit us to obtain a reliable quantifiable marker of breathing effort (see comment below for more details). This is now indicated in the limitations on page 22.

"... since the MR respiratory sensor used cannot quantify respiratory effort"

Pg 7 end of para 1: What sensor was used?

A respiratory sensor MR compatible (Philips Healthcare, Best, The Netherlands) that can capture respiratory motion with an inflated cushion positioned just below the sternum. This produces a qualitative output that is adequate to determine inspiration and expiration, but the signal is affected by participant movement and position, so cannot be used reliably to quantify breathing across the whole scanning session.

This is now indicated on page 7.

*“A respiratory sensor **MR compatible (Philips Healthcare, Best, The Netherlands)** that can capture respiratory motion with an inflated cushion positioned just below the sternum was used to align the tagged images with the respiratory cycle.”*

Pg 9: Were swallows 'dry' or induced by a liquid bolus?

The swallows were 'dry' as participants were not given anything liquid to swallow. The manuscript has been amended on page 9:

“ Therefore, each intramuscular wire was assigned to one of the four tongue neuromuscular compartments based on the EMG recorded during ‘dry’ swallowing, as oblique and horizontal compartments have distinct motoneuronal outputs: monophasic for the horizontal and biphasic for the oblique (Yeung et al., 2022). ”

Referee #2:

Jugé et al present a novel study describing regional genioglossus movement and neural activity levels in OSA patients, providing unique insight into reduced neuromuscular efficiency (failure to translate genioglossus activity into improved mechanics) that is a cause of OSA in many individuals. The study incorporates a rare combination of functional imaging and intramuscular EMG with pharyngeal pressure measurements. These physiological data are unique and highly valuable for our field, particularly given the highly ambitious and nature of the data collection. Investigators demonstrated commitment to rigor, with a notable example being the rescoring of sleep study data from outside their sites. Below I have detailed concerns that I feel could be addressed to even further raise the quality of the work. There are several ways modern statistical methods could be employed that I believe would have a notable impact on the findings.

1) The study is a descriptive physiological study that describes regional heterogeneity in phasic genioglossus activation and accompanying inspiratory movement (or the surprising lack therefore) across compartments. The first key message is that genioglossus activity (EMG) does not consistently translate into forward movement, even during wakefulness. I believe this message should come first in the abstract, and be supported by quantitative results describing how greater GG activity levels were not associated with increased movement levels.

We have clarified this message in the 4th key points summary “ **OSA participants had a ~7-times higher odds of having at least one compartment for which a larger anterior tongue movement was not associated with a higher phasic EMG than controls. Therefore, higher genioglossus phasic EMG might not consistently translate into tongue dilatory movement, particularly in people with OSA**”.

Due to the word limit, this was not included in the abstract, but we believe the current reporting of the results is sufficient to indicate it.

“ Only for 106 compartments (68%) a larger anterior (dilatory) movement was associated with a higher phasic EMG (mixed linear regression, Beta = 0.089, 95% CI [0.000, 0.178], t(99) = 1.995, P = 0.049, hereafter EMG \uparrow /mvt \uparrow). For the remaining 50 (32%) compartments, a larger dilatory movement was associated with a lower phasic EMG (mixed linear regression, Beta = -0.123, 95% CI [-0.224, -0.022], t(43) = -2.458, P = 0.018, hereafter EMG \downarrow /mvt \uparrow). ”

Thank you for the suggestion.

Currently the authors have used partial correlation analysis in Results, but ideally this analysis would involve a linear mixed model analysis describing anterior movement as a function of genioglossus activity (fixed effect) and subject (random effect) to take proper advantage of the repeated measurement design. The authors will still be expected to find that there is no strong association between movement and genioglossus activity in general (see also 6 below for adjustments for additional confounders).

Thank you for this useful suggestion. In the revised manuscript, mixed linear regression analyses were used to assess the association between tongue dilatory movement (dependent variable) and EMG measurements and their interaction (fixed effects) while controlling for repeated measures within the same participant (random effects) and adjusting for nadir epiglottic pressure.

Abstract, methods (statistical analysis), results and Figure 4 were amended accordingly.

This new analysis strategy did not change results except for the overall relationship between regional tongue movement and EMG activity. In the initial analysis, the associations were not significant. In contrast, with the mixed linear regression analysis, we found a significant relationship between movement and phasic EMG, and two significant interactions (i.e., phasic EMG*nadir epiglottic pressure and tonic EMG*nadir epiglottic pressure). Please see Table 4 on page 15 for more information. SPSS output is included below for the reviewer's information.

Estimates of Fixed Effects^a

Parameter	Estimate	Std. Error	df	t	Sig.	95% Confidence Interval	
						Lower Bound	Upper Bound
Intercept	.368	.225	149	1.635	.104	-.077	.812
Tonic_EMG	-.096	.051	149.000	-1.885	.061	-.196	.005
Pha_EMG	.061	.029	149	2.074	.040	.003	.118
Tonic_EMG * Pha_EMG	-.002	.003	149	-.699	.486	-.007	.003
P_nadir	.091	.060	149.000	1.505	.134	-.028	.210
Pha_EMG * P_nadir	.013	.006	149.000	2.296	.023	.002	.024
Tonic_EMG * P_nadir	-.040	.012	149.000	-3.366	<.001	-.064	-.017

a. Dependent Variable: corr_mvt_positive.

We interpret this latter result as indicating that for a given phasic and tonic EMG, tongue compartments in participants with less negative nadir epiglottic pressure are associated with smaller anterior dilatory movement. Therefore, overall regional tongue movements do not directly correlate with genioglossus activity in all cases unless baseline anatomical compromise is controlled for, for which we use nadir epiglottic pressure as a proxy here.

2) To me, the second key physiological insight is that is an association between forward genioglossus movement and EMG in some compartments but not others, which is a novel and important message.

a. First, the analysis needed for the definition of "EMG↗/mvt↗" is overly abbreviated in Methods and needs clarification.

This subgroup of "EMG↗/mvt↗" appears to be defined based on "groups 1 and 2" in Results but should defined more precisely in Methods and again in the appropriate Figure legend for clarity.

Otherwise, reader may interpret these incorrectly as (1) a simple anterior dilatory movement (since phasic EMG is always positive), or (2) a positive within-breath time-series correlation between EMG and movement synchronized with the respiratory cycle (note that the term "correlation" is used in the definition but no strict correlation is performed for the definition).

More precise definitions have been added to the methods (statistical analysis) and the legend of Figure 4 (end of the manuscript on page 34):

*"To describe the observed relationships between tongue dilatory movement and phasic EMG, we used the following terminology $EMG \nearrow / mvt \nearrow$ when a higher phasic EMG was associated with a larger tongue dilatory movement **for the neuromuscular tongue compartments classified as having either a "low" activity with minimal movement or "high" activity with large movement**, and $EMG \searrow / mvt \nearrow$ when a lower phasic EMG was associated with a larger tongue dilatory movement (i.e., decoupled relationship) **for the neuromuscular tongue compartments classified as having a "low" activity with large movement or a "high" activity with minimal anterior movement.**"*

*" Figure 4. . (A) For 106 neuromuscular tongue compartments **classified as having either a "low" activity with minimal anterior movement (>1mm) or "high" activity with large anterior movement (<1mm)**, a larger anterior (dilatory) tongue movement was associated with higher phasic EMG (beta = 0.089, 95% confidence intervals [0.000, 0.178], $t(99) = 1.995$, $P = 0.049$). (B) For the remaining compartments (n = 50) **classified as having either a "low" activity with large anterior movement or a "high" activity with minimal anterior movement**, a larger dilatory movement was associated with a lower phasic EMG (beta = -0.123, 95% confidence intervals [-0.224, -0.022], $t(43) = -2.458$, $P = 0.018$). Tonic EMG was not related to anterior tongue movement in both cases. **Mixed linear regressions accounting for the fact that up to 4 measurements were made from each participant (random effects) were used to assess associations between tongue dilatory movement (dependent variable) and EMG measurements and their interaction (fixed effects) with the covariate nadir epiglottic pressure (Table 5). Simple linear regression 95% with confident intervals are plotted and the equation reported on each graph. Note: positive displacement indicates anterior (dilatory) movement and "low" and "high" levels of activity are defined in Table 3.***

Please also replace $EMG \searrow / mvt \nearrow$ with $EMG \nearrow / mvt \searrow$; presumably there is no drop in EMG per se (i.e. negative phasic activity), rather the movement is considered the dependent variable in this analysis. Instead, of $EMG \nearrow / mvt \searrow$ the authors could consider using "efficient"/"inefficient" as shorthand for the reduced neuromuscular efficiency these compartments can be concluded to exhibit.

We respectfully prefer not to change the terminology for the decoupled relationship (i.e., $EMG \searrow / mvt \nearrow$) because we believe it reflects better the unexpected aspect of relationship observed in Figure 4B, i.e. a larger anterior dilatory movement was associated with a lower phasic EMG. If we had only observed that an increase in EMG did not result in a greater movement, this could be explained by a more negative pharyngeal pressure counteracting the dilatory force from genioglossus, which would be less unexpected. We agree that this relationship does not mean *per se* that the phasic EMG dropped, but we interpret this result as an indication that greater dilatory motion could occur without an increase in regional genioglossus activity, making the genioglossus tissue more compliant and, thus, more able to deform in response to activation of other tongue muscles. If verified, this would also mean this relationship is not mechanically "inefficient", but rather decoupled of the expected relationship (i.e., $EMG \nearrow / mvt \nearrow$).

Key point summary #4 has been amended (page 2):

“ ... a larger anterior tongue movement was not associated with a higher phasic EMG “

Discussion, 1st paragraph, page 22:

*“In those tongue compartments with a decoupled EMG↘/mvt↗ relationship, greater dilatory motion might occur without increased in regional genioglossus activity. **A possible mechanism could be that the lower muscle activity makes** that region more compliant, and thus more able to deform in response to activation of other tongue muscles.”*

b. Re: "For 106 of the compartments (68%), a larger anterior (dilatory) movement was associated with a higher phasic EMG ($r = -0.26$, $P = 0.01$, hereafter EMG↗/mvt↗). In contrast, for the remaining 50 (32%) compartments, a larger dilatory movement was associated with a lower phasic EMG ($r = 0.49$, $P < 0.001$, hereafter EMG↘/mvt↗)." As these groups are defined by their EMG↘/mvt↗ status, the direction of the correlation appears to be 'by definition' positive and negative in magnitude; this means that the main quantitative parameter of interest is how many compartments are in each group. I believe it is fine to report the R-values (and 95CI in R if desired) but I would remove this from the Abstract, and certainly remove p-values throughout as the p-values do not capture how these groups were selected using EMG and directional movement behavior (that are then used within the correlation). I don't believe this hurts the work in any way.

Good point. We agree. We have therefore opted to report the results of the mixed linear analysis instead (comment #1), and we have clarified how these relationships were obtained from the identification of 4 groups where needed in the manuscript. We have also opted to report the simple linear regression equation in Figure 4 for each group to describe those relationships further.

3) Perhaps the most important key message made by the authors is that, compared to controls, OSA patients appear to have a higher proportion of compartments with inefficient (EMG↗/mvt↘) translation from GG to anterior movement. However, the analysis performed is not sufficiently powerful. With the data available in Figure 5, a preliminary analysis using multinomial ordinal logistic regression revealed that there is a meaningful association between OSA status and a higher likelihood of an increased number of inefficient (EMG↗/mvt↘) compartments, odds ratio [95%CI]: 6.2 [1.12-35] ($p=0.037$). The authors should perform a similar analysis in place of the Fisher Exact test, and consider including the finding in the abstract.

We thank the reviewer for their enthusiasm in conducting this analysis, and after discussing with our statistical consultant, we have undertaken a variant of the suggested analysis that also adjusts for epiglottic pressure and has greater statistical power. i.e. A binary logistic regression analysis adjusting for nadir epiglottic pressure showed that OSA participants had a higher odds of having at least one decoupled EMG↘/mvt↗ compartment (odds ratio [95%CI]: 7.53 [1.19 - 47.47] ($P = 0.032$)). This result has been added to the main body of the manuscript (page 17), the abstract and key point summary (pages 2 and 3) and the captions of Figure 5 (page 34).

Following on from this suggestion, we also conducted a similar analysis for the data in Figure 6, which plots the number of decoupled compartments against movement pattern. First, for each participant we dichotomised the movement pattern into minimal vs other and grouped the compartment data into whether there was one or more decoupled compartment. We then computed a binary logistic regression while adjusting for nadir epiglottic pressure (odds ratio [95%CI]: 1.47 [0.948 – 19.91] ($P = 0.059$)), which did not find a significant results. This is reported in the main body of the manuscript 9 page 17) and the captions of the Figure 6, page 35.

4) As noted by the authors, the two-way ANOVA analysis of whether compartment type or OSA status explains genioglossus activity (Figure 2) does not take advantage of the repeated measures across subjects, and also lacks power and flexibility. Linear mixed effects models analysis would be more appropriate here and would account for the duplicate measures within a compartment and the missing compartment data. There appears to be a signal for increased activity and posterior compartments that would be of interest. (First, model GG as a function of compartment type as a fixed effect, subject as a random effect; Second, add OSA status to the first model; Third add additional adjustment for confounders, see 6 below)

Thank you for this suggestion. In consultation with our biostatistician, we have computed a mixed linear regression to assess the effect of OSA, tongue compartments, and their interactions (fixed effects) on phasic and tonic EMG (dependent variables) while adjusting for nadir epiglottic pressure. Results are reported in the manuscript on page 12, Figure 2 and Table 2.

“ ... for phasic and tonic EMG, no significant effect of OSA status and tongue compartments were found, and there was no significant interaction between OSA status and tongue compartment”.

The EMG activity is uniform.

5) "Upper airway function" is used in place of "upper airway movement" throughout the manuscript; immediate concern is that the movement is being measured in most analyses herein. The authors will be aware that absence of movement may not imply failure of function depending on the pressures/forces that the region of interest is being exposed to. e.g. higher EMG could fail to generate anterior movement but be successfully increasing forces and preventing posterior movement in the presence of exposure to larger negative pressures; function might be normal but movement would be absent. Can the authors be more judicious with their use of "function" versus "movement" in interpretation where the totality of their data supports reduced anterior movement alone or when unadjusted for pressures.

Fair point. "Upper airway function" has been changed to "upper airway movement" as appropriate throughout the manuscript.

Eg. Conclusions: " This has important implications for upper airway control as it demonstrates that upper airway muscle movement cannot be directly inferred from EMG measurements alone and vice versa, ..."

6) To provide further data supporting use of 'function' more so that 'movement' alone: While it is clear that OSA patients have a higher likelihood of inefficient GG compartments, it is not entirely clear that the failure to translate GG into movement can not be at least in part attributable to greater baseline anatomical compromise; adjusting analysis for Pepi swing magnitude and end-expiratory cross sectional area would help to address this and enable authors to more strongly make conclusions based on function. I strongly recommend adding these variables in sensitivity analysis as confounders to the mixed model and ordinal regression analyses recommended above. For example, it is plausible that the analysis recommended in (1) above would demonstrate that, after adjusting for these confounders, greater GG activity is associated with greater anterior (i.e. less posterior) movement. It is also plausible that the analysis recommended in (3) above would demonstrate an attenuation of the odds for greater number of inefficient compartments when these confounders are accounted for. If so we might learn that OSA is associated with greater inefficiency per se, or is associated with greater inefficiency in part via a greater baseline anatomical compromise. Both

findings are of interest.

As mentioned above in response to previous comments, we have amended our analysis to adjust for the nadir epiglottic pressure. A limitation of our work is that our airway cross-sectional area measurements represent the average size across the whole respiratory cycle as they were measured on anatomical images collected over multiple respiratory cycles. As a result, we cannot also adjust our analysis with the end-expiratory upper airway cross-sectional area.

Once our analysis was adjusted, we found that 1- people with less negative nadir epiglottic pressure move their less tongue for a given level of EMG activity (comment #1), and 2- an increase in the odds of 'decoupled' compartments in OSA participants (odds ratio [95%CI]: **5.75** [1.04 - 31.67] (P = 0.032) vs odds ratio [95%CI]: **7.53** [1.19 - 47.47] (P = 0.032) when adjusted) (comment #3).

Taken together, this suggests that as a result of baseline anatomical compromise, OSA participants had a greater risk of decoupling between inspiratory tongue movement and genioglossus EMG activity. This could be a successful strategy to achieve a greater tongue anterior dilatory movement since participants with less negative nadir epiglottic pressure had less movement for a given EMG activity, but this remains to be proven.

This is now discussed in the first two paragraphs of the "Associations between regional inspiratory tongue dilatory movement and genioglossus EMG activity" section in the discussion from page 21.

*"Airway patency is preserved during wakefulness in people with OSA via active stiffening and dilatory movement of the tongue, with the magnitude and spatial extent of this motion being greater in people with more severe OSA (Juge et al., 2020a). However, how much the tongue can move to dilate the airway is influenced by both neural drive and local mechanics (Bilston & Gandevia, 2014). In this study, people with OSA had a larger tongue volume and more negative nadir epiglottic pressure than control participants, and **the overall relationship between genioglossus EMG activity and anterior tongue dilatory movement suggests that people with less negative nadir epiglottic pressure move their tongue less during inspiration for a given activity (Table 4)**. A closer examination of the data also established that the genioglossus inspiratory activity and dilatory tongue movement relationship was decoupled (EMG↘/mvt↗) in almost one-third (32%) of the neuromuscular compartments. **OSA participants had a ~7-times higher odds of having at least one decoupled EMG↘/mvt↗ compartment than non-OSA participants, which may contribute to OSA pathogenesis.***

*"Change in tongue shape can be achieved by various different combinations of muscle contraction since it is a muscular hydrostat (Kier & Smith, 1985). In those tongue compartments with a decoupled EMG↘/mvt↗ relationship, greater dilatory motion might occur **without an increased regional regional genioglossus activity. A possible mechanism could be that the lower muscle activity makes that region more compliant, and thus, more able to deform in response to activation of other tongue muscles.** This could represent an alternative strategy to maintain patency in a particularly confined oral cavity.... "*

Minor

Figure 2. Can the y-axes be made to have the same magnitude for consistency? Tonic EMG has a larger y-range.

Done.

Is it confusing to have the "A-P movement" variable quantitatively described by negative values for the anterior direction. Can you rename to "horizontal posterior movement" or similar so that the defined direction is not the exact reverse of the intuitive direction? Even so, it will still remain somewhat challenging that the "mvt \nearrow " describes increased anterior movement but values shown are negative. Perhaps just make the anterior values positive to solve all the issues here?

Anterior values of the movements have been changed to positive.

I believe Mathworks is in Natick rather than Danvers.

Correct. The manuscript has been amended.

Prefer the use of genioglossus "activity" rather than genioglossus "drive"

The manuscript has been amended accordingly.

E.g. Conclusions: *" Relationships between genioglossus activity and tongue movement during wakefulness are complex and vary between individuals."*

Re: "However, it is unclear why some are unable to activate their tongue muscles to maintain airway patency during sleep (Younes, 2008)." Consider also citing Gell et al 2022 Thorax who provided some supplemental data on patients with neuromechanical inefficiency if you think it is appropriate.

Thank you for the suggestion. The citation was added to the manuscript on page 4.

Figure 3 legend. Accounting-> accounted (switch to consistent past tense).

Done.

AHI definition was not provided; can the authors comment on the criteria (AHI3pa vs AHI4)?

Hypopneas were scored per AASM definition (3% desaturation). The definition has been added to the manuscript page 6 (last paragraph) :

" Standard diagnostic polysomnography was conducted within a year of the recruitment to determine the apnea-hypopnoea index (AHI) (American Academy of Sleep Medicine criteria v2.4 (3% desaturation) (Berry et al., 2017)),..."

END OF COMMENTS

Dear Dr Bilston,

Re: JP-RP-2023-285187R1 "Regional association between inspiratory tongue dilatory movement and genioglossus activity during wakefulness in people with obstructive sleep apnoea" by Lauriane Jugé, Angela Liao, Jade Yeung, Fiona Knapman, Christopher Bull, Peter GR Burke, Elizabeth C. Brown, Simon C Gandevia, Danny J Eckert, Jane E Butler, and Lynne E Bilston

We are pleased to tell you that your paper has been accepted for publication in The Journal of Physiology.

Authors should note that it is too late at this point to offer corrections prior to proofing. The accepted version will be published online, ahead of the copy edited and typeset version being made available. Major corrections at proof stage, such as changes to figures, will be referred to the Editors for approval before they can be incorporated. Only minor changes, such as to style and consistency, should be made at proof stage. Changes that need to be made after proof stage will usually require a formal correction notice.

Yours sincerely,

Scott K. Powers
Senior Editor
The Journal of Physiology
<https://jp.msubmit.net>
<http://jp.physoc.org>
The Physiological Society
Hodgkin Huxley House
30 Farringdon Lane
London, EC1R 3AW
UK
<http://www.physoc.org>
<http://journals.physoc.org>

P.S. - You can help your research get the attention it deserves! Check out Wiley's free Promotion Guide for best-practice recommendations for promoting your work at www.wileyauthors.com/eeo/guide. You can learn more about Wiley Editing Services which offers professional video, design, and writing services to create shareable video abstracts, infographics, conference posters, lay summaries, and research news stories for your research at www.wileyauthors.com/eeo/promotion.

IMPORTANT NOTICE ABOUT OPEN ACCESS: To assist authors whose funding agencies mandate public access to published research findings sooner than 12 months after publication, The Journal of Physiology allows authors to pay an Open Access (OA) fee to have their papers made freely available immediately on publication.

You can check if your funder or institution has a Wiley Open Access Account here: <https://authorservices.wiley.com/author-resources/Journal-Authors/licensing-and-open-access/open-access/author-compliance-tool.html>.

EDITOR COMMENTS

Reviewing Editor:

Thank you for this thoughtful revision of a strong physiological study revealing new information about an important clinical problem. There are some very minor editorial suggestions from the referee you may want to consider.

Senior Editor:

Congratulations on the completion of an outstanding study. Thank you for submitting your work to the Journal of Physiology.

REFeree COMMENTS

Referee #1:

Thank you for addressing the previous minor points I raised. I commend the authors on a very interesting and important study further revealing the complexity of airway control in obstructive sleep apnoea. The observations have important implications for the field, both prospectively and retrospectively.

Referee #2:

The authors have been highly responsive to both reviewers. The recommendations were carefully addressed and led to useful improvements. The manuscript makes an important contribution to the field.

Minor suggested edits that the authors can be trusted to make if they choose: 1) "a higher odd of" -> "greater odds of" in Abstract. 2) "binary logistic regression" can just be "logistic regression" after first mention or throughout (safe to assume binary based on context).

1st Confidential Review

25-Sep-2023